# Apico-basal cell compression regulates Lamin A/C levels in epithelial tissues

K. Venkatesan Iyer [1,2,6 ✉], Anna Taubenberger[3], Salma Ahmed Zeidan[1], Natalie A. Dye[1,4], Suzanne Eaton[1,4,7] & Frank Jülicher [2,4,5 ✉]

The levels of nuclear protein Lamin A/C are crucial for nuclear mechanotransduction. Lamin A/C levels are known to scale with tissue stiffness and extracellular matrix levels in mesenchymal tissues. But in epithelial tissues, where cells lack a strong interaction with the extracellular matrix, it is unclear how Lamin A/C is regulated. Here, we show in epithelial tissues that Lamin A/C levels scale with apico-basal cell compression, independent of tissue stiffness. Using genetic perturbations in *Drosophila* epithelial tissues, we show that apico-basal cell compression regulates the levels of Lamin A/C by deforming the nucleus. Further, in mammalian epithelial cells, we show that nuclear deformation regulates Lamin A/C levels by modulating the levels of phosphorylation of Lamin A/C at Serine 22, a target for Lamin A/C degradation. Taken together, our results reveal a mechanism of Lamin A/C regulation which could provide key insights for understanding nuclear mechanotransduction in epithelial tissues.

[1] Max Planck Institute of Molecular Cell Biology and Genetics, Dresden, Germany. [2] Max Planck Institute for the Physics of Complex Systems, Dresden, Germany. [3] Biotechnology Center TU Dresden, Dresden, Germany. [4] Cluster of Excellence Physics of Life, TU Dresden, Dresden, Germany. [5] Center for Systems Biology Dresden, Dresden, Germany. [6] Present address: Department of Mechanical Engineering, Indian Institute of Science, Bangalore, India. [7] Deceased: Suzanne Eaton. ✉email: iyer@mpi-cbg.de; julicher@pks.mpg.de

Mechanotransduction—a process through which mechanical forces are converted to biochemical signaling or gene expression is crucial for physiology[1–3]. Impaired mechanotransduction is at the heart of various diseases[1]. Mechanotransduction could be activated by either extrinsic forces like fluid shear flow in blood vessels[4] or intrinsic forces through actomyosin contractility[5]. Recent experiments have highlighted the importance of the transmission of forces to the nucleus through the cytoskeleton[6–8]. In this process, the nuclear scaffold plays an important role in activating mechanotransduction in the nucleus[9].

Nuclear Lamins are the primary component of the nuclear scaffold. Lamins are type V intermediate filaments[10], comprising of A-type and B-type Lamins. In vertebrates, A-type Lamins are composed of two splice isoforms of the *Lamin A* gene, Lamin A, and Lamin C, whereas B-Type Lamins include Lamin B1 and Lamin B2. In contrast, in invertebrates like *Drosophila*, Lamin C is the only form of A-type Lamin, and Lamin DM$_0$ is the only type of B-type Lamin[11]. Lamins are known to influence the mechanical properties of the nucleus. Lamin A/C contributes to the stiffness and viscosity of the nucleus whereas Lamin B is responsible for the elasticity of the nucleus[12,13]. Recent experiments have shown that levels of Lamin A/C are crucial for mechanotransduction in the nucleus[14]. Not only does Lamin influence force-induced stiffening of the nucleus[15–17], it also promotes, nuclear translocation of MKL, a co-factor of Serum Response factor (SRF)[18]. Interestingly, Lamin B is expressed uniformly in all tissues during development, but Lamin A/C is developmentally regulated and has tissue-specific expression profiles[19].

Over the last decade, studies have focused on identifying how Lamin A/C is regulated in different tissues. Experiments in mesenchymal stem cells (MSC), mesenchymal tissues and some non-mesenchymal tissues have shown that Lamin A/C scales with extracellular matrix (ECM) stiffness in these tissues[20]. Epithelial tissues are one of the most abundant adult tissues. ECM in these tissues is scant[21], but the cells adhere to each other through a plethora of cell-cell junctions along the apico-basal axis[22]. These cell–cell junctions bear most of the mechanical stress in the tissue[23]. In the absence of strong interactions with the ECM, it is still unclear how Lamin A/C levels are regulated in these tissues. Studying this would be key to understand the role of Lamin A/C in mechanotransduction and the interplay between tissue mechanics and nuclear mechanotransduction in epithelial tissues.

In this work, using *Drosophila* epithelial tissues and mammalian Madin Darby Canine Kidney (MDCK) cells as model systems, we provide strong evidence that apico-basal cell compression is an ECM-independent mechanism for regulation of Lamin A/C in epithelial tissues. Nuclear deformation in response to apico-basal cell compression modulates Lamin A/C levels. By combining genetic perturbations in vivo, and altered cell packing, in cultured mammalian cells, we show that apico-basal cell compression-based regulation of Lamin A/C is evolutionarily conserved in epithelial tissues.

## Results

**Lamin C levels vary across different epithelial tissues**. As a first step toward investigating how Lamin A/C is regulated in epithelial tissues, we measured the levels of Lamin A/C in epithelial tissues. We dissected Salivary gland[24], trachea[25], and wing disc[26] epithelial tissues from late third instar *Drosophila* larvae (Fig. 1a) and immunostained them using an antibody against Lamin C (LamC), the only isoform of Lamin A/C expressed in *Drosophila*. We estimated the levels of nuclear LamC by measuring the mean

pixel intensity of LamC fluorescence staining from a maximum intensity projection of a z-stack images (Fig. 1l and Methods). We tested that epitope masking[27] does not affect our staining of LamC (Supplementary Fig. 1, see Methods). We found that LamC levels are low in the nuclei of salivary gland (SG) cells (Fig. 1b, c) and highest in the nuclei of tracheal cells (Fig. 1d, e). We observed significant differences of LamC levels even within a single wing disc. When we measured the levels of LamC in different regions of the wing disc we found that LamC is very low in the wing pouch (Fig. 1f–k). As compared to pouch cells, LamC is about three-fold higher in the fold regions and about four-fold higher in the PM cells (Fig. 1f–k). Upon comparing LamC level in the SG and trachea with the wing disc pouch, we observed that the level of LamC in the SG is similar to that of the wing pouch, whereas tracheal cells have about five-fold higher levels of LamC (Fig. 1m).

In order to test whether these differences in levels of LamC across tissues are specific to LamC, we immunostained these tissues with an antibody against Lamin DM$_0$, an ortholog of Lamin B in *Drosophila* that is known to be expressed throughout development[19]. Interestingly, we found no significant differences in the levels of LamDM$_0$ between different *Drosophila* tissues (Supplementary Fig. 2a–l). Moreover, the levels of another nuclear envelope associated LINC complex protein, Klaroid (Koi), also did not vary between different regions of the wing disc (Supplementary Fig 2m, n). These results indicate that epithelial tissues have different levels of LamC and that the observed differences are not an artifact of immunostaining or deep tissue imaging.

**Lamin C scales with apico-basal cell compression in epithelial tissues**. Next, we investigated the origin of the differences in LamC levels across *Drosophila* epithelial tissues. A characteristic feature of epithelial tissues is their cell packing density, defined as the number of cells per mm$^2$ surface area of the tissue. Squamous epithelial tissues have a very low packing density, whereas pseudostratified tissues have a very high packing density. As a consequence of the low packing density in squamous tissues, cells are compressed apico-basally. We examined whether LamC could scale with the cell packing density or apico-basal compression of cells. To do so, we measured the apical cell area ($A$) and apico-basal height ($h$) in SG, trachea, and different regions of the wing disc (Fig. 2a–e). In order to measure apico-basal compression of cells, we approximated the cells in the epithelia as cuboids and defined for each cell an apico-basal cell compression index (Fig. 2f)

$$D_{\text{cell}} = \ln\left(\frac{A}{h^2}\right) \tag{1}$$

We defined $D_{\text{cell}}$ such that it is close to 0 for cuboidal cells, negative for columnar cells that are elongated along the apico-basal axis, and positive for squamous cells, compressed along the apico-basal axis. In accordance with previous studies[28], we found that for the SG, both apical area and apico-basal height are very large, resulting in a $D_{\text{cell}}$ close to 1. In contrast, both the trachea and PM have large apical areas but very small apico-basal cell heights, leading to large values of $D_{\text{cell}}$ (Fig. 2g and Supplementary Fig. 3a, b), as shown previously[29,30]. In the wing disc pouch, the apical area is small, but the apico-basal height is large, resulting in a small $D_{\text{cell}}$. The $D_{\text{cell}}$ values averaged within a tissue, varied widely among different *Drosophila* tissues (Fig. 2g). In order to observe the possible influence of $D_{\text{cell}}$ on LamC levels in *Drosophila* tissues, we plotted the levels of LamC in these tissues against $D_{\text{cell}}$ of individual cells in different tissues (Fig. 2h). We found that LamC levels showed a non-linear

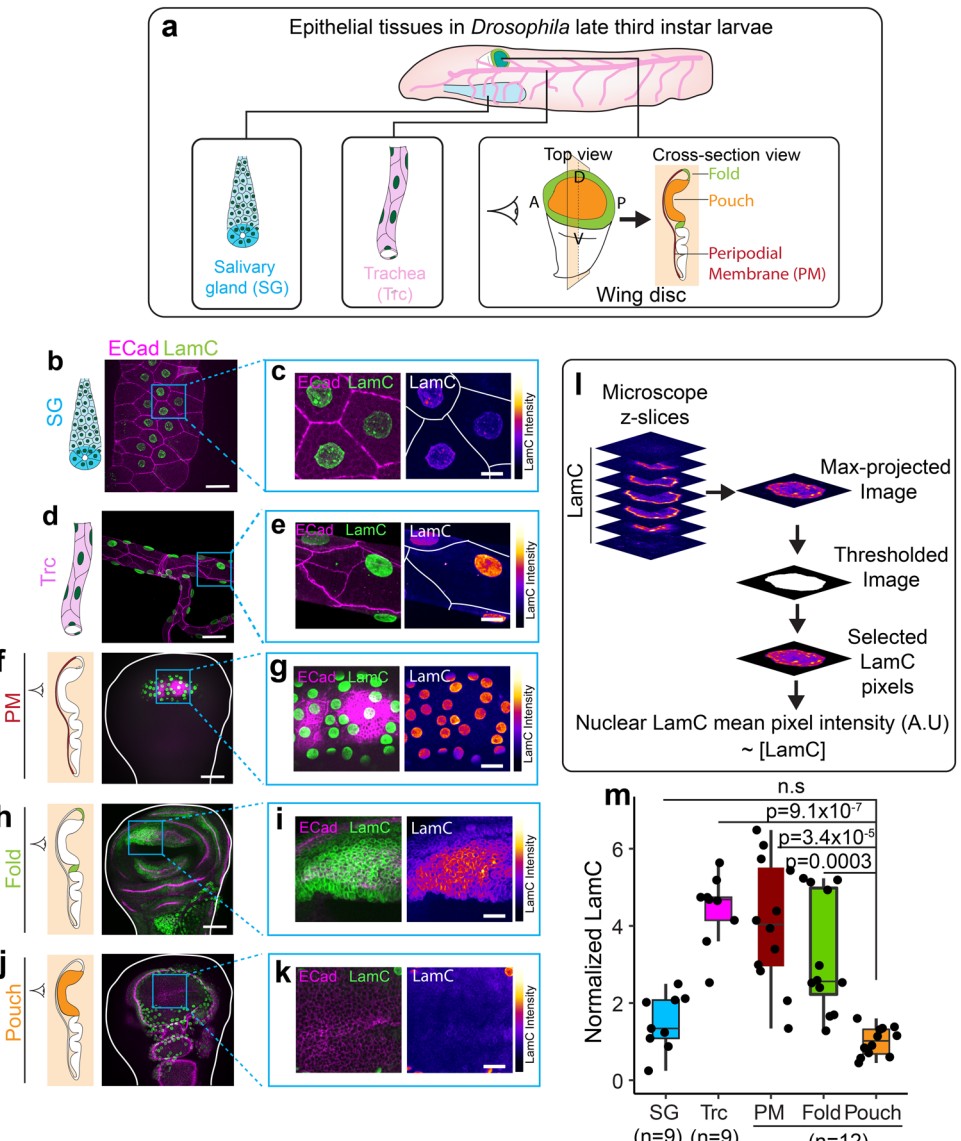

**Fig. 1 LamC levels vary across different epithelial tissues. a** Schematic showing different epithelial tissues in *Drosophila*. **b** LamC in salivary glands. Image of trachea showing E-Cad (magenta) and LamC (green). **c** Enlarged image of the region marked by the blue ROI in (**b**). Left panel shows the merge of E-Cad and LamC. Right panel shows the color-coded LamC intensity. Solid white lines represent cell boundaries. **d** LamC in trachea. Image of salivary gland showing E-Cad (magenta) and LamC (green). **e** Enlarged image of the region marked by the blue ROI in (**e**). Left panel shows the merge of E-Cad and LamC. Right panel shows the color-coded LamC intensity. Solid white lines represent cell boundaries. **f** LamC in trachea. Image of PM showing E-Cad (magenta) and LamC (green). **g** Enlarged image of the region marked by the blue ROI in (**f**). Left panel shows the merge of E-Cad and LamC. Right panel shows the color-coded LamC intensity. **h** LamC in the fold. Image of fold showing E-Cad (magenta) and LamC (green). **i** Enlarged image of the region marked by the blue ROI in (**h**). Left panel shows the merge of E-Cad and LamC. Right panel shows the color-coded LamC intensity. **j** LamC in the pouch. Image of pouch showing E-Cad (magenta) and LamC (green). **k** Enlarged image of the region marked by the blue ROI in (**j**). Left panel shows the merge of E-Cad and LamC. Right panel shows the color-coded LamC intensity. **l** Schematic showing the estimation of LamC levels in the nucleus of epithelial tissues. **m** Box-plot showing the normalized levels of LamC in the epithelial tissues. Normalization is performed with respect to the levels in the pouch. The sample number, *n* represents the number of biologically independent tissue samples. One-way ANOVA was performed to estimate *p* values. The *p* values are shown in comparison to LamC levels in the pouch. *P* values are indicated in the respective plots and n.s represents the differences are not significant. Scale bar in overview images (**b**, **d**, **f**, **h**, **j**), 25 μm. Scale bar in enlarged images, 15 μm. In the box-plot, horizontal line represents the median of the data, lower and upper bounds of the box represent the 25th and 75th percentile of data, and the whiskers represent the minimum and maximum of the data. The scattered point on the box represents the actual data points.

correlation with $D_{cell}$, with a Spearman's rank correlation of 0.72. We observed a similar correlation between LamC and $D_{cell}$ when averaged over multiple cells in a given tissue (Supplementary Fig 3c). These results show that LamC in epithelial tissue scales with the apico-basal cell compression.

**LamC levels do not correlate with epithelial tissue stiffness and ECM levels**. Since LamC levels in mesenchymal and non-epithelial tissues are known to scale with tissue stiffness and matrix rigidity[20,31], we investigated whether such scaling exists also for epithelial tissues. Towards this end, we probed the stiffness

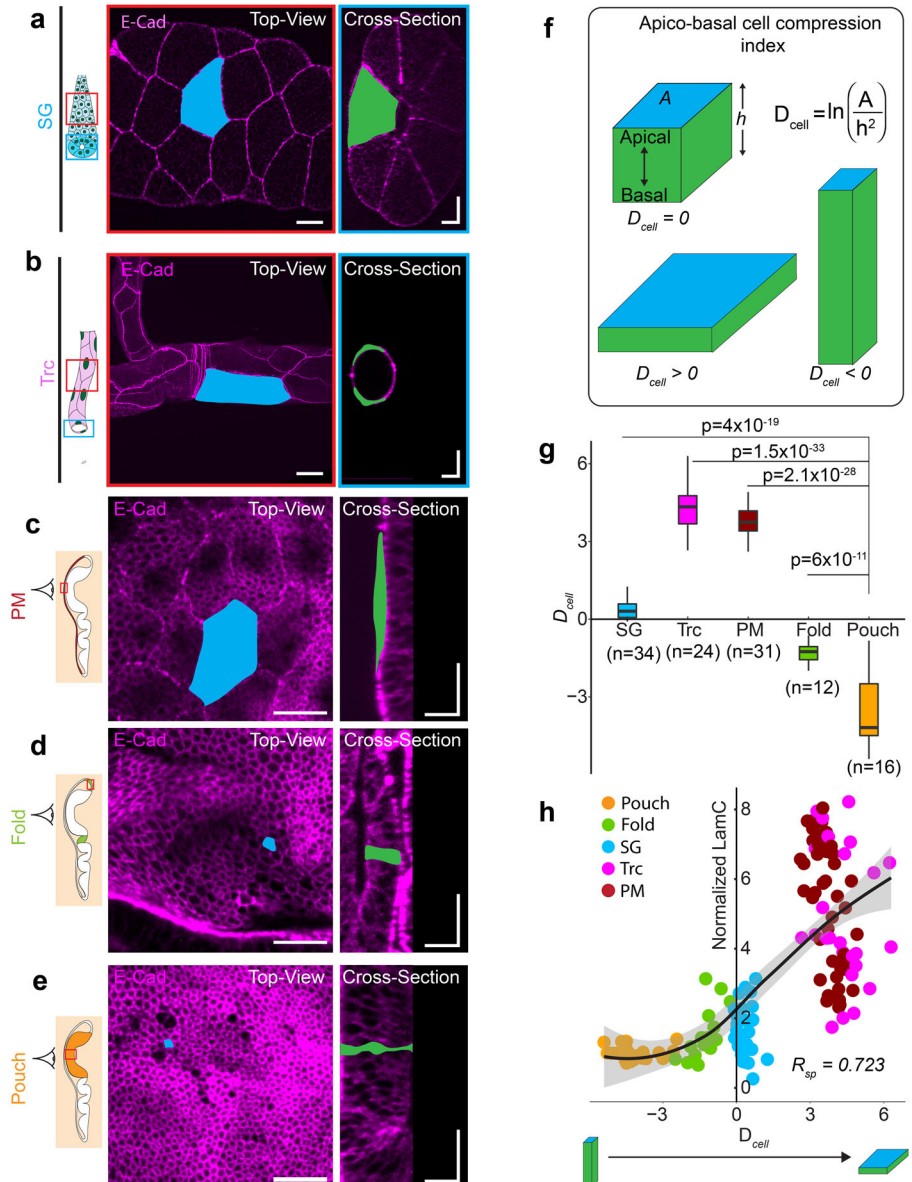

**Fig. 2 Lamin C scales with apico-basal cell compression in epithelial tissues.** Schematic representation of the salivary gland (**a**) and trachea (**b**) with the region imaged marked by the red and blue ROIs. Images show the top view (red ROI) and cross-section view (blue ROI) of the SG and trachea expressing E-Cad$^{GFP}$ (magenta). A representative cell is shown with apical area marked in blue and apico-basal area marked in green. Scale bar in top-view images in (**a**) and (**b**), 25 μm. Scalebar in cross-section images, 25 μm along both axes. **c–e** Cell morphology in different regions of the wing disc. PM of the wing disc is colored in dark red (**c**). The fold is colored in light green (**d**), and the pouch is colored in orange (**e**). Images show the top view and cross-section view of each wing disc region expressing E-Cad$^{GFP}$ (magenta). A representative cell is shown with the apical area marked in blue and apico-basal area marked in green. Scale bar in top-view images, 15 μm. Scale bar in cross-section images, 15 μm along both axes. **f** Schematic representation of cell deformation index ($D_{cell}$) for epithelial tissues. $D_{cell}$ for a cube is 1, whereas $D_{cell}$ for a cuboid elongated along the apico-basal axis is <1 and a cuboid compressed along the apico-basal axis is >1. **g** Box-plot of $D_{cell}$ for different epithelial tissues of *Drosophila*. The sample number, $n$ represents the number of cells analyzed across 10 different tissue samples. **h** Scatter plot between $D_{cell}$ and normalized levels of LamC. Each point in the scatter plot represents an individual cell in different *Drosophila* tissues. Solid black line represents the LOESS regression of the data. The Spearman rank correlation coefficient, $R_{sp}$ is 0.723. P values are estimated by one-way ANOVA and the $p$ values are shown for comparison with the $D_{cell}$ value in the pouch.

of the basal surface of the epithelial tissues using atomic force microscopy (AFM) indentation experiments (Fig. 3a). We observed a large variation in stiffness among different epithelial tissues. The apparent Young's moduli detected for salivary gland and trachea were between 15 and 20 kPa, whereas the apparent Young's moduli in different regions of the wing disc varied between 1 and 5 kPa (Fig. 3b). Interestingly, unlike previous studies in mesenchymal tissues, stiffness of epithelial tissues showed no significant correlation with LamC levels (Fig. 3c). Next,

we investigated whether LamC levels are regulated by the levels of ECM. Towards this end, we measured three different ECM proteins—Collagen IV, Laminin A, and Perlecan in trachea, salivary gland, and wing disc by measuring the mean fluorescence intensity (intensity/pixel) of an endogenously tagged Collagen IV (vkg$^{GFP}$), Laminin-A (LanA$^{GFP}$), and Perlecan (Trol$^{GFP}$) from maximum intensity projected images. We found that in these epithelial tissues, the levels of collagen IV are much larger than Laminin-A and Perlecan, suggesting that Collagen IV is a major component of

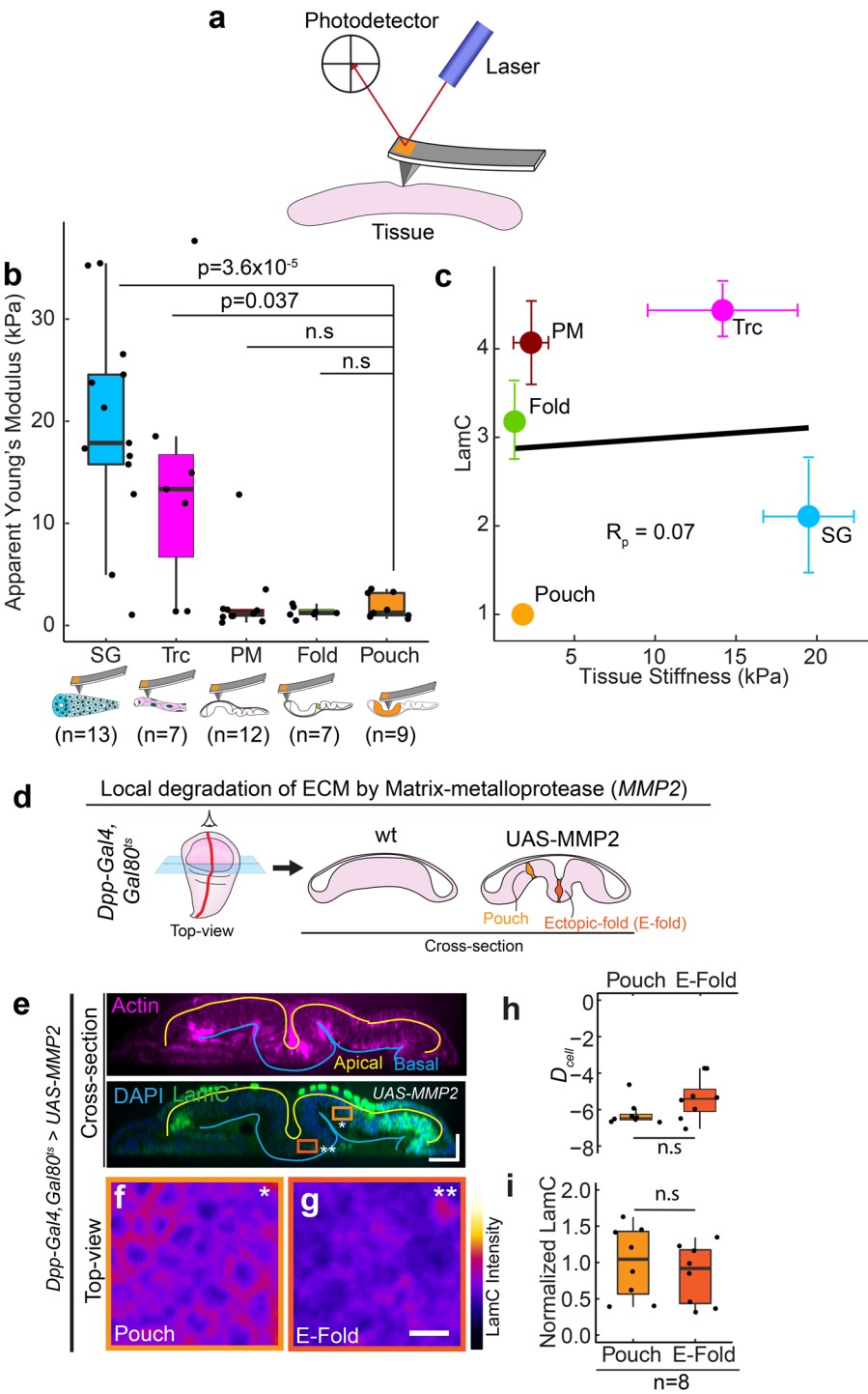

ECM in these tissues (Supplementary Fig. 4). Interestingly, when we plotted the levels of LamC against the levels of Collagen IV across different epithelial tissues, we did not find any significant correlation between them (Supplementary Fig. 5a) (Pearson Correlation coefficient, −0.57 with False Discovery Rate of 0.18). These experiments show that unlike mesenchymal tissues, LamC levels in epithelial tissues do not correlate with levels of major ECM components in epithelial tissues.

Next, we investigated whether ECM is required for maintaining the LamC levels in the wing disc. To this end, we used Matrix metalloprotease to degrade the ECM[32]. We locally degraded the ECM in the wing disc by expressing *Drosophila* matrix

metalloprotease (MMP2) in a spatially confined stripe using *Dpp-GAL4* (Fig. 3d, e). Degrading the ECM by expressing *UAS-MMP2* in the Dpp stripe is known to induce an ectopic fold in the wing disc[33]. We degraded the ECM for 24 h in the late L3 stage of larval development by using a temperature-sensitive Gal80 (*Gal80ts*). Temporal control is crucial, as prolonged ECM degradation is lethal for the larva. Upon degrading the ECM (visualized by Collagen IV) an ectopic fold was formed (Fig. 3e and Supplementary Fig 5c), and we observed a small decrease in cell height, in accordance with previous studies[33,34] (Supplementary Fig. 7a). The cell shape index ($D_{cell}$), however, did not change significantly between *wt* and *UAS-MMP2* overexpression (Fig. 3h).

**Fig. 3 LamC levels do not correlate with epithelial tissue stiffness and ECM levels. a** Schematic showing the measurement of tissue stiffness using atomic force microscope indentation. **b** Box-plot showing tissue stiffness (in kPa) for different *Drosophila* tissues. The sample number *n* represents the number of biologically independent tissue samples analyzed. **c** Scatter plot between tissue stiffness and LamC levels in epithelial tissues. Each point represents the average value of tissue stiffness and LamC levels in a tissue. Solid black line shows the linear fit to the data. Pearson's correlation coefficient, $R_p$ is 0.07. **d** Schematic showing the local degradation of ECM by overexpression of MMP2 using Dpp-Gal4. The left panel shows the top-view of the wing disc and the right panel shows the cross-section view of the wt wing disc and wing disc expressing UAS-MMP2. **e** Cross-section through the DV boundary of the wing disc expressing *UAS*-MMP2 by Dpp-Gal4. The upper panel shows actin (magenta), and the lower panel shows the merge between LamC and DAPI. Solid yellow line shows the apical surface of cells, and the blue line shows the basal surface. Scale bar along both axes, 25 μm. **f, g** Color-coded LamC images of the regions shown in (**b**). The pouch region is shown by the light orange ROI and single asterisk, and the ectopic fold (E-fold) region is shown by the dark orange ROI and double asterisk. Scale bar, 5 μm. **h** Box-plot showing $D_{cell}$ in wing pouch and E-fold. **i** Box-plot showing normalized LamC in the pouch and the E-fold. The sample number, *n* represents the number of independent tissue samples. Normalization is done with respect to the pouch. The *p* values between two samples are estimated with a two-sided Student's *t*-test and between multiple samples by one-way ANOVA. The *p* values in (**b**) are shown in comparison to the levels of collagen IV in the wing pouch. *P* values are indicated in the respective plots and n.s represents that the differences are not significant. In the box-plot, horizontal line represents the median of the data, lower and upper bounds of the box represent the 25th and 75th percentile of data, and the whiskers represent the minimum and maximum of the data. The scattered point on the box represents the actual data points.

Interestingly, LamC levels were also not significantly different between pouch and ectopic fold (Fig. 3f, g and i), consistent with our observation that LamC levels correlates with $D_{cell}$. Collagen crosslinking has been shown to influence the stiffness of the ECM[35]. In order to test the role of Collagen crosslinking on LamC levels, we knocked down Peroxidasin—a protein that crosslinks Collagen, in a stripe of cells in the wing disc expressing *Dpp-Gal4, Gal80$^{ts}$*. We observed no changes in the morphology of the cells and LamC levels as compared to the cells outside the stripe (Supplementary Fig. 5d). Though previous experiments have shown that interaction of the cells with ECM is necessary to maintain cell shape in the wing disc[34], our experiments show that LamC levels in wing disc epithelia are not regulated by levels and crosslinking of ECM proteins.

**Apico-basal cell compression regulates Lamin C levels in epithelial tissues.** Since the levels of LamC correlate strongly with apico-basal cell compression in *Drosophila* epithelial tissues, we hypothesized that LamC levels are regulated by apico-basal cell compression. To test this hypothesis, we modulated cell shapes in the wing discs and observed its effect on LamC levels. First, we reduced the apico-basal height of the cells by overexpressing a dominant-negative form of CDC42 ($CDC42^{F89}$) in a stripe using *Dpp-GAL4* (Fig. 4a, b). CDC42 has been shown previously to regulate the apico-basal elongation of cells in the wing pouch[36]. Overexpression of $CDC42^{F89}$ altered the organization of actin filaments without influencing the rate of proliferation of cells (Supplementary Fig. 6). Consistent with previous studies[36], this induced a deep ectopic fold by reducing cell height and increasing the apical cross-section area of cells (Supplementary Fig. 7c, d). This perturbation resulted in a significant increase in $D_{cell}$ of the cells in the ectopic fold as compared to the cells outside the fold (Fig. 4e), suggesting that the cells in the ectopic fold are apico-basally compressed. In response to this elevated cell compression in the ectopic fold, we observed a significant increase in LamC levels as compared to the cells outside the ectopic fold (Fig. 4c, d, f). Next, we increased cell height by ectopically overexpressing a transcription factor *Lines* (*Lin*) in the PM using Ubx-Gal4 (Fig. 4g). Usually, *Lin* is expressed in the pouch but is absent in the PM. Upon overexpressing *Lin* in the PM, we observed a significant increase in cell height, as shown previously[37,38] (Fig. 4h–k and Supplementary Fig. 6e, f). We also observed that the cells overexpressing *Lin* have a reduced apical area as compared to *wt* (Supplementary Fig. 6). When we measured $D_{cell}$ for PM in *wt* and *Lin* overexpressing cells, we observed that $D_{cell}$ is significantly lower in the PM of *Lin* overexpressing cells (Fig. 4l), suggesting that apico-basal cell compression is reduced in these

cells. In response to this reduced cell compression, LamC levels in the *Lin* overexpressing PM cells were significantly lower than the PM cells of wt wing discs (Fig. 4m). Moreover, in the *Lin* over-expressed wing disc, $D_{cell}$ values and the levels of LamC between the pouch and the PM were indistinguishable (Supplementary Fig. 7g, h). Interestingly, when we plotted the cell deformation index, $D_{cell}$ and normalized LamC for all the perturbations, all these data followed similar correlation shown in Fig. 2h (Supplementary Fig. 7i), suggesting that the levels of Lamin A/C can be predicted solely from the apico-basal deformation of cells. These results support our hypothesis that LamC levels are regulated by apico-basal compression of cells. However, they do not rule out the possibility that a correlation between cell compression and LamC levels exists because changes in LamC levels influence cell shapes.

To test this possibility, we directly perturbed the levels of LamC and observed its influence on cell shapes. We first downregulated LamC in the PM of the wing disc, by driving LamC RNAi using Ubx-Gal4, which is primarily expressed in the PM. We found that in wing discs expressing LamC RNAi, LamC levels are significantly reduced as compared to wild-type (Supplementary Fig. 8a, c). But, the apical cell area and height between wild-type and LamC RNAi knockdown cells were indistinguishable (Supplementary Fig. 8b, d). Next, we overexpressed LamC in a stripe of cells using Ptc-Gal4. We restricted expression to the late L3 larval stage by using the temperature-sensitive *Gal80$^{ts}$* and shifting the larvae from 25 °C to 29 °C (Supplementary Fig. 8e). This perturbation resulted in significantly higher levels of LamC in cells along the stripe (Supplementary Fig. 8f). However, the height of the cells in the stripe was identical to the cells outside the stripe (Supplementary Fig. 8g, h). Even in LamC mutant larvae, the height of cells in the wing pouch were identical to the cells in the wild-type wing disc (Supplementary Fig 8i). Taken together, these results provide compelling evidence for a unidirectional coupling between cell mechanics and LamC regulation in epithelial tissues—apico-basal compression of epithelial cells regulates LamC, but altering LamC does not influence cell morphology in epithelial tissues.

Recent studies have shown that Yorkie (yki) is essential for maintaining the flattened morphology of epithelia in *Drosophila* follicle cells and peripodial membrane of the wing disc[39]. Since flattening of cells is necessary for inducing LamC levels, we tested whether overexpression of yki could flatten the pouch cells and induce LamC levels. Towards this, we expressed either *UAS-yki$^{GFP}$* or a hyperactivated form of yki, *UAS-yki$^{S168A}$* in a stripe of cells in the wing pouch using *Dpp-GAL4*. We found that overexpression of either *yki$^{GFP}$* of *yki-S168A* does not induce a change in the cell morphology and LamC levels in the wing pouch

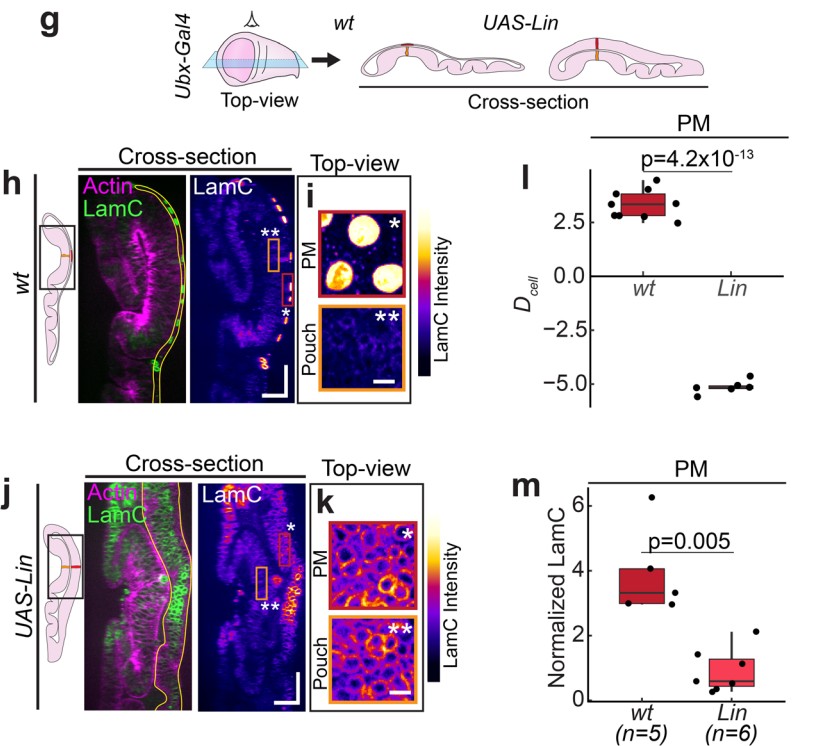

(Supplementary Fig. 9). This shows that though Yorkie is important for maintaining the flat morphology of epithelial cells, it does not induce apico-basal compression and elevate LamC levels in the wing disc.

**Apico-basal cell compression regulates LamC via prolate to oblate deformation of the nucleus**. Next, we investigated whether apico-basal compression of cells induces nuclear deformation leading to increase in Lamin A/C levels. We visualized nuclear shape in Salivary glands, trachea, and wing discs (Fig. 5 and Supplementary Fig. 10a, b). We measured cross-sectional area

$(A_n)$ and nuclear height $(h_n)$ in these nuclei. We defined the nuclear deformation index as

$$D_{\text{nuc}} = \ln\left(\frac{4A_n}{\pi h_n^2}\right) \qquad (2)$$

We defined $D_{\text{nuc}}$ such that $D_{\text{nuc}} = 0$ for a spherical nucleus and deviates from 0 as the nuclear shape deviates away from a sphere. $D_{\text{nuc}}$ is negative for prolate shapes and $D_{\text{nuc}}$ is positive for oblate shapes. The nuclei of Salivary glands have a large cross-sectional area and nuclear height (Supplementary Fig. 10a, b), making $D_{\text{nuc}}$

**Fig. 4 Apico-basal cell compression regulates LamC in epithelial tissues. a** Schematic showing the local reduction in cell height by overexpression of a dominant-negative form of CDC42 using Dpp-Gal4. Left panel shows the top-view of the wing disc and the right panel shows the cross-section view of the wing disc. **b** Cross-section through the DV boundary of the wing disc expressing CDC42$^{F89}$ by Dpp-Gal4 along the AP boundary. Upper panel shows actin (magenta) and lower panel shows merge between LamC and DAPI. Solid yellow line shows the apical surface, and the blue line shows the basal surface of the cells. Scale bar along both axes, 25 μm. **c**, **d** Color-coded LamC images of the regions shown in (**b**). The pouch region is shown by the light orange ROI and single asterisk, and the ectopic fold (E-fold) region is shown by the dark-orange ROI and double asterisk. Scale bar, 5 μm. **e** Box-plot showing $D_{cell}$ in the wing pouch and E-fold. **f** Box-plot showing normalized LamC in pouch and the E-fold. Normalization is done with respect to pouch. The sample number, $n$ represents the number of wing discs analyzed. **g** Schematic showing the increase in cell height of PM cells by ectopic overexpression of *UAS-Lin* using Ubx-Gal4. Left panel shows the top-view of the wing disc and the right panel shows the cross-section view of the wing disc. **h** Cross-section of the *wt* wing disc in the region of the wing disc shown by the black ROI. Left panel shows merge of actin (magenta) and Lam C (green). Solid yellow line shows the PM. Right panel shows the color-coded image of LamC. Scale bar along both axes, 25 μm. **i** Top-view color-coded LamC intensity of the PM (marked by the dark-red ROI and single asterisk) and pouch (light-orange ROI and double asterisk). Scale bar, 5 μm. **j** Cross-section images of wing disc with ectopic overexpression of *UAS-Lin* in the region shown by black ROI. Left panel shows merge of actin (magenta) and Lam C (green). Solid yellow line shows the PM. Right panel shows the color-coded image of LamC. Scale bar along both axes, 25 μm. **k** Top-view color-coded LamC intensity of the PM (marked by the dark-red ROI and single asterisk) and pouch (light-orange ROI and double asterisk). Scale bar, 5 μm. **l** Box-plot showing $D_{cell}$ for PM in wt and upon UAS-Lin overexpression. **m** Box-plot showing normalized LamC for PM in *wt* and upon UAS-Lin overexpression. The sample number $n$ represents the number of wing discs analyzed. The $p$ values are estimated using a two-sided Student's $t$-test and the corresponding $p$ values are given in plots. In the box-plot, horizontal line represents the median of the data, lower and upper bounds of the box represent the 25th and 75th percentile of data, and the whiskers represent the minimum and maximum of the data. The scattered point on the box represents the actual data points.

close to 0. In contrast, trachea has a significantly larger $D_{nuc}$, indicating that nuclei are oblate in trachea as compared to salivary glands (Fig. 5f). Measuring nuclear shape in the wing disc is challenging because segmentation of the densely packed nuclei in the disc pouch is difficult. We overcame this problem by driving a nuclear-localized GFP called "Stinger-GFP" with *Ubx-Gal4* (Fig. 5a). This Gal4 primarily expresses in the PM but has a patchy expression in the wing pouch (Supplementary Fig 10c). Thus, nuclei in the wing pouch and fold are sparsely labeled, enabling nuclear segmentation (Fig. 5a). When we measured the cross-sectional area and height of the nuclei in the pouch and fold (Supplementary Fig 10d, e), we found that the nuclei have a very low $D_{nuc}$ and they are prolate (Fig. 5f). The nuclei in the PM have very large $D_{nuc}$ and they are oblate. Interestingly, $D_{nuc}$ and $D_{cell}$ in *Drosophila* tissues are strongly correlated, with a correlation coefficient of 0.93 (Fig. 5g).

We observed a significant correlation between cell shape and nuclear shape. We next asked whether changes in nuclear shape could lead to changes in the levels of LamC. To answer this, we expressed a dominant-negative form of CDC42 (*CDC42$^{L89}$*) in a thin stripe of cells in the wing disc using *Dpp-GAL4* (Fig. 5h, i) and measured nuclear shape using high-resolution imaging of these wing discs. We measured the cross-sectional area and height of the nucleus, both in the unperturbed pouch cells and the cells in the ectopic fold (Fig. 5h, i). The nuclei in the pouch remained slightly prolate whereas the nuclei in the ectopic fold had oblate shapes (Fig. 5h, i), as follows from the $D_{nuc}$ values for the pouch and the ectopic fold nuclei (Fig. 5p). Concomitantly, we observed that the LamC levels were increased in the ectopic fold, as compared to the perturbed pouch where they remained low (Fig. 5q). These results suggest that the observed changes in LamC in *Drosophila* tissues are caused by prolate to oblate nuclear shape change in response to apico-basal cell compression.

**Regulation of Lamin A/C by apico-basal cell compression is also observed in mammalian epithelial cells**. Next, we asked whether such apico-basal cell compression-dependent LamC regulation is conserved in mammalian cells. To this end, we cultured mammalian Madin Darby Canine Kidney (MDCK) epithelial cells on a cover glass coated with Collagen I. In order to mimic the cell packing in the wing disc pouch and PM, we cultured the cells either at high density (~5700 cells/mm$^2$) or at low density (~70 cells/mm$^2$) (Fig. 6a–d). We cultured both low−density and high-density cultures on the same culture dish

coated with Collagen I to provide an ECM-independent micro-environment. We observed that the apical area of the cells grown at high density was much smaller than that grown at low density (Fig. 6a–d). Concomitantly, the apico-basal height of cells grown at higher density was significantly larger than those grown at low density (Supplementary Fig 10f). We found that the nuclear deformation index, $D_{nuc}$ is strongly correlated with cell deformation index, $D_{cell}$, with a correlation coefficient of 0.97 (Fig. 5l). This is consistent with previous studies showing that nuclear shape correlates with cell shape in cells in culture[40–42]. These results in cultured MDCK cells show that nuclear deformation by the apico-basal cell compression is an evolutionarily conserved phenomenon in epithelial tissues.

Next, we visualized Lamin A/C in MDCK cells by immunostaining the cells with an antibody against Lamin A (LamA)—a canine ortholog of *Drosophila* LamC. We used an antibody that has been shown previously to be not influenced by epitope masking[27] (Supplementary Fig. 11a–d, see Methods). Using this antibody, we performed the staining of both low-density and high-density cultures in the same culture dish, to minimize variation in the levels of LamA in low-density and high-density cultures. We found that the levels of LamA were significantly higher in cells plated at low density as compared to cells plated at high density (Fig. 6f). We validated this result by performing Western blot on cells cultured in low and high density. We found a twofold increase in LamA in low density culture, consistent with the immunofluorescence data (Supplementary Fig. 11e, f). Interestingly, the levels of LamB1 were similar in LD and HD cells (Supplementary Fig. 11g, h). The mean level and total amount of LamA in the nucleus show a strong and non-linear correlation with $D_{nuc}$ (Spearman's rank correlation = 0.86) (Fig. 6g and Supplementary Fig. 10g). This sigmoidal relationship between LamA and $D_{nuc}$ suggests the existence of a threshold in the response of Lamin A/C to shape change of the nucleus, similar to the threshold in nuclear envelope unfolding shown in recent studies[43,44] These results suggest that regulation of Lamin A/C by apico-basal compression of cells is evolutionarily conserved, in epithelial tissues.

**Lamin network stretching induced dephosphorylation of Lamin A/C regulates Lamin A/C levels in epithelia**. Lamin A/C is highly phosphorylated during mitosis[45]. The phosphorylation of Lamin A/C at serine 22 (pSer22) is a well-studied phosphorylation site[45]. It is known that during mitosis pSer22 results in

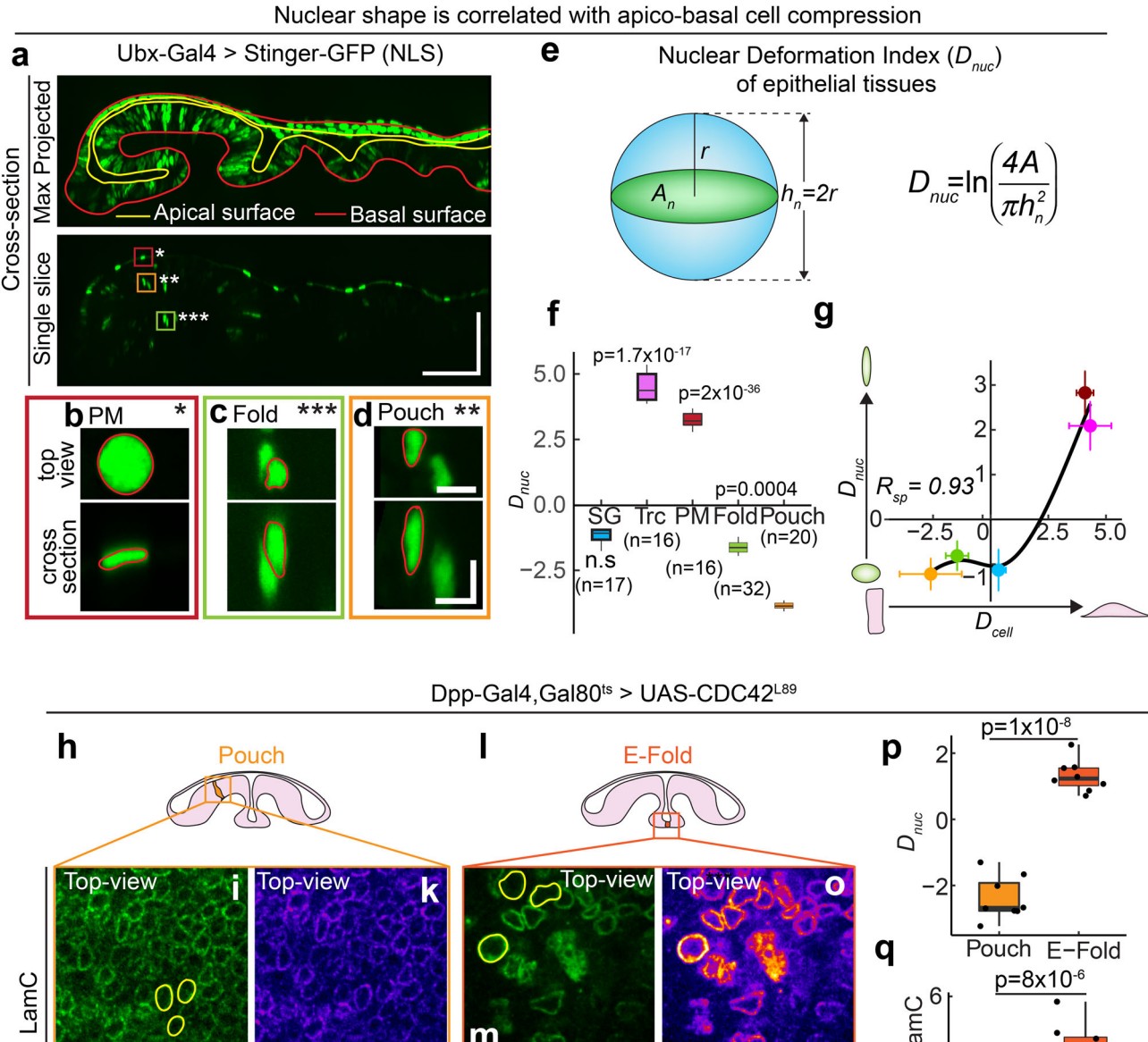

degradation of Lamin A/C[46]. Recent experiments have shown that pSer22 levels during interphase depend on the mechanical tension on nuclear lamina[46]. We therefore asked whether stretching of Lamina due to nuclear flattening could result in the reduction of pSer22 levels and hence higher levels of Lamin A/C.

To address this question, we first measured the surface area strain of the nucleus, an indicator of stretching of the Lamin network. Epithelial cells in suspension attain a spherical morphology with a spherical nucleus. Since, a sphere has the smallest surface area for a given volume, any deformation from a sphere would result in an increase in the surface area. During nuclear flattening, the nucleus transitions from a sphere to an oblate spheroid. We measured the surface area of the oblate

spheroid ($S_{oblate}$) and calculated the change in surface area from that of a sphere ($S_{sp}$) of the same volume (see Methods). The surface area strain ($\Delta S$) is given by (Fig. 7a)

$$\Delta S = \frac{S_{oblate} - S_{sp}}{S_{sp}} \tag{3}$$

Surface area strain is significantly higher in low-density cells as compared to high-density cells, suggesting that the Lamina of low-density cells is more stretched than high-density cells. Consistent with this observation, Lamin A/C in the nuclei of HD cells is wrinkled, whereas no wrinkles are observed in the

**Fig. 5 Apico-basal cell compression regulates Lamin A/C levels via prolate to oblate deformation of the nucleus. a** Cross-section view of the wing disc expressing nuclear NLS protein Stinger-GFP driven by Ubx-Gal4. Upper panel shows the maximum projected image of Stinger-GFP. Solid red line shows the basal surface of cells, and solid yellow line shows the apical surface of cells. Scale bar along both axes, 50 μm. Enlarged images of regions where PM is marked by dark-red ROI and single asterisk (**b**), fold by green ROI and triple asterisk (**c**), and pouch by orange ROI and double asterisk (**d**). For each region, the upper panel shows the top-view of the nucleus, and lower panel shows the cross-section of the nucleus. Solid red line shows the outline of the nucleus. Scale bar along both axes, 5 μm. **e** Schematic showing the estimation of nuclear deformation index ($D_{nuc}$) for a nucleus. **f** Box-plot showing $D_{nuc}$ for different *Drosophila* tissues. The sample number, $n$ represents the number of nuclei analyzed across 10 tissue samples. **g** Scatter plot between $D_{cell}$ and $D_{nuc}$ in *Drosophila* tissues—SG (light blue), Trc (Magenta), PM (dark red), Fold (light green) and Pouch (orange). Data are presented as mean ± S.D. The Spearman's rank correlation coefficient, $R_{sp}$ is 0.93 with FDR of 0.049. Solid black line represents the LOESS regression to the data. **h** Schematic showing an elongated cell in the wing pouch. **i** XY-view of the LamC image. Equatorial area of three representative nuclei is marked by yellow outlines. **j** Cross-section view of LamC in pouch. YZ cross-sectional area of three representative nuclei is marked by yellow outlines. **k** Color-coded image showing the intensity of LamC in the pouch nuclei. **l** Schematic showing an apico-basally compressed cell in the ectopic fold. **m** XY-view of the LamC image. Equatorial area of three representative nuclei is marked by yellow outlines. **n** Cross-section view of LamC in pouch. YZ cross-sectional area of three representative nuclei are marked by yellow outlines. **o** Color-coded image showing the intensity of LamC in the pouch nuclei. Scalebar, 10 μm in vertical and horizontal directions. **p** Box-plot showing $D_{nuc}$ for pouch and E-fold nuclei. **q** Box-plot showing the normalized LamC levels in Pouch and E-fold nuclei. Normalization is performed against LamC levels in the pouch. Sample number $n$ represents the number of wing discs analyzed. The $p$ values are estimated by two-sided Student's $t$-test and the $p$ values are indicated in the respective plots. $P$ values in (**f**) are shown for comparison with the $D_{nuc}$ for the pouch. In the box-plot, horizontal line represents the median of the data, lower and upper bounds of the box represent the 25th and 75th percentile of data, and the whiskers represent the minimum and maximum of the data. The scattered point on the box represents the actual data points. The scattered point on the box represents the actual data points.

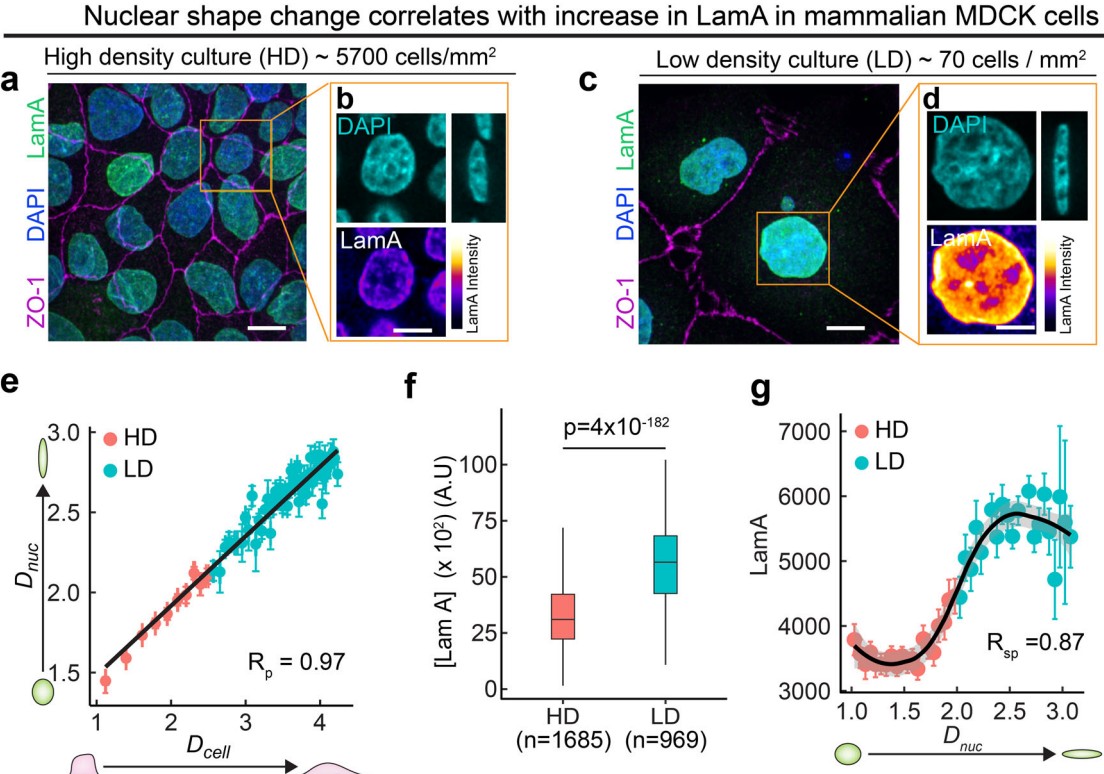

**Fig. 6 Regulation of Lamin A/C by apico-basal cell compression is also observed in mammalian epithelial cells. a** Image showing high-density culture of 5700 cells/mm². Tight junction protein ZO-1 is shown in magenta, Lamin A in green, and DAPI in blue. Scalebar, 15 μm. **b** Enlarged images of the region shown by the orange ROI in (**h**). DAPI is shown in blue and Lamin A is shown in color-coded intensity. Scale bar, 10 μm. **c** Image showing low density culture of 70 cells/ mm². Tight junction protein ZO-1 is shown in magenta, Lamin A in green, and DAPI in blue. Scalebar, 15 μm. **d** Enlarged images of the region shown by orange ROI in (**j**). DAPI is shown in blue, and Lamin A is shown in color-coded intensity. Scale bar, 10 μm. **e** Binned scatter plot between $D_{cell}$ and $D_{nuc}$ generated by binning $D_{cell}$ in bins of 0.07. Each point in the plot represents mean value of $D_{cell}$ and $D_{nuc}$ in the bin. The red points represent data from HD culture and cyan points represent data from LD culture. The data are presented as mean ± S.E.M. Solid red line represents the linear regression to the data. Pearson's correlation coefficient $R_p$ is 0.97. **f** Box-plot showing LamA in high density (HD) and low density (LD) cultures. The sample number, $n$ represents the number of cells analyzed over 3 independent biological replicates. **g** Binned scatter plot between $D_{nuc}$ and LamA intensity generated by binning $D_{nuc}$ in bins of 0.06. Each point in the plot represents mean value of $D_{nuc}$ and LamA in the bin. The red points represent data from HD culture and cyan points represent data from LD culture. The data are presented as mean ± S.E.M. Solid black line represents the LOESS regression to the data. Spearman's rank correlation coefficient $R_{sp}$ is 0.87 In the box-plot, horizontal line represents the median of the data, lower and upper bounds of the box represent the 25th and 75th percentile of data, and the whiskers represent the minimum and maximum of the data. The scattered point on the box represents the actual data points. $p$ values are estimated by two-sided Student's $t$-test and the $p$ values are indicated in the respective plots.

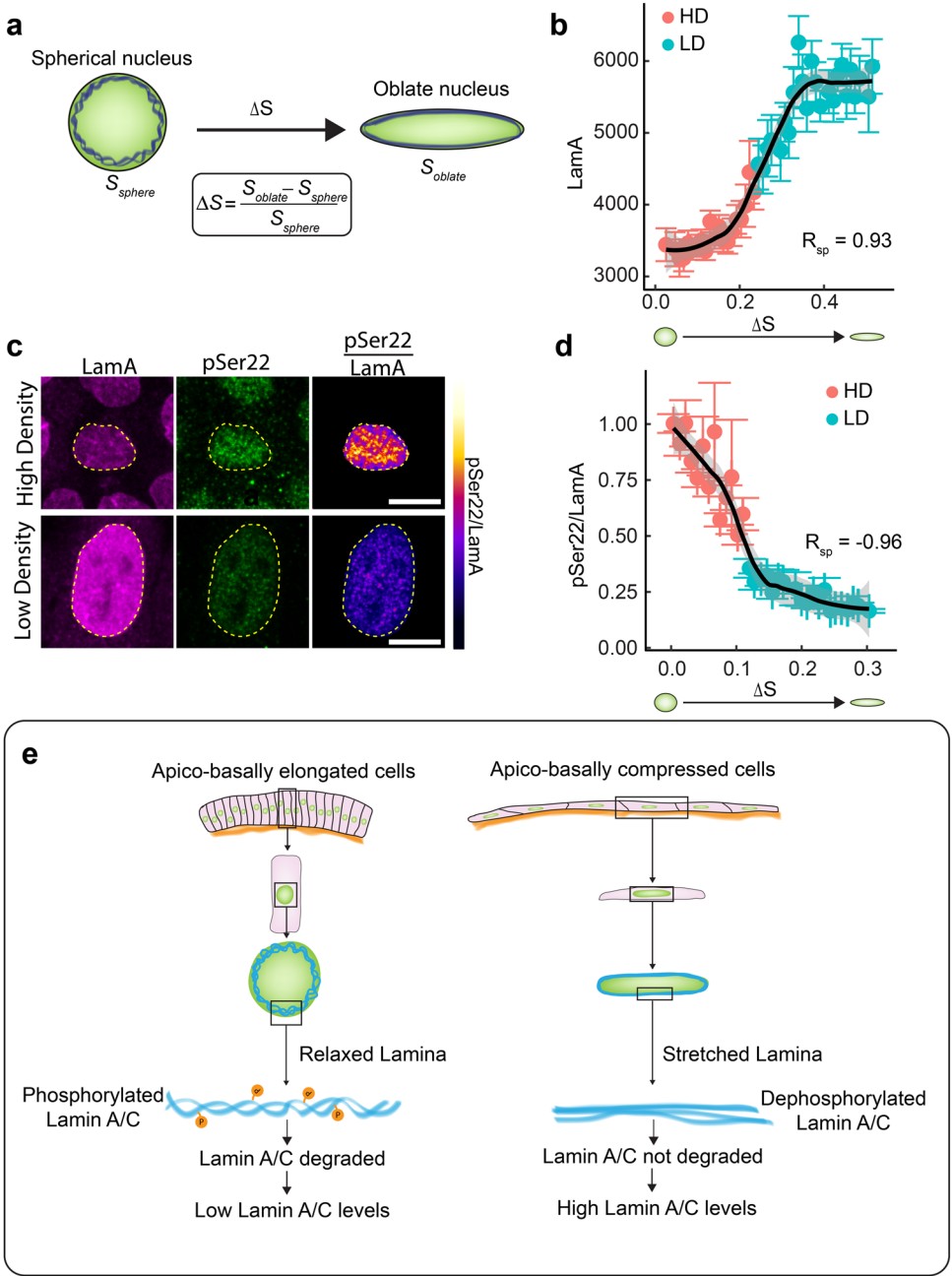

**Fig. 7 Lamina network stretching induced dephosphorylation of Lamin A/C regulates Lamin A/C levels in epithelial tissues. a** Schematic showing the stretching of nuclear lamina (blue) when a sphere is compressed to an oblate spheroid. The surface area strain, represented by ΔS is the change in surface area of the oblate spheroid as compared to the surface area of the sphere. **b** Binned scatter plot between Surface area strain (ΔS) and Lamin A generated by binning ΔS in bins of 0.01. Each point in the plot represents the mean value of ΔS and LamA in the bin. The red points represent data from HD culture and cyan points represent data from LD culture. Solid black line shows the LOESS regression to the data. Spearman's rank correlation coefficient between ΔS and LamA $R_{sp}$ is 0.93. **c** Images showing Lamin A(magenta), pSer22 (green) and pSer22/LamA ratio (color-coded) in cells plated in high density (top panel) or low density (bottom panel). Scale bar, 10 μm. **d** Binned scatter plot between surface area strain (ΔS) and pSer22/LamA generated by binning ΔS in bins of 0.01. Each point in the plot represents the mean value of ΔS and pSer22/LamA in the bin. The red points represent data from HD culture and cyan points represent data from LD culture. Spearman's rank correlation coefficient between ΔS and pSer22/LamA, $R_{sp}$ is −0.93. Solid black line shows the LOESS regression to the data. Data are presented as mean values ± S.E.M. **e** A model elaborating the apico-basal cell compression mediated regulation of Lamin A/C in epithelial tissues.

nuclei of LD cells (Supplementary Fig. 11e, f). When we plotted LamA levels as a function of ΔS, we observed a strong and non-linear correlation with a Spearman's rank correlation coefficient of 0.93 (Fig. 7b). We observed similar correlation between surface area strain and LamC in *Drosophila* tissues as well (Supplementary Fig. 11c, d). This result shows that a threshold surface strain

is required to alter the levels of Lam A in cells. In order to understand whether this non-linear relationship between surface strain and LamA stems from the dephosphorylation of LamA, we stained the cells with antibodies against LamA and pSer22 (Fig. 7c). We estimated the fraction of phosphorylated LamA by measuring the ratio of pSer22 to LamA. We observed that though

LamA levels are lower in HD cells as compared to LD cells, the fraction of phosphorylated LamA is significantly higher in HD cells as compared to LD cells (Supplementary Fig. 11a). We observed a strong non-linear dependence of the fraction of phosphorylated LamA, on surface area strain and $D_{nuc}$ (Fig. 7d and Supplementary Fig. 11b), indicating that the fraction of phosphorylated LamA is reduced upon stretching of the Lamin network. These results clearly show that increasing surface area strain reduces the phosphorylation of LamA, resulting in a reduced degradation of Lamin A. This change could lead to the higher levels of LamA in LD cells with an oblate nucleus and large surface area strain (Fig. 7e).

## Discussion

Lamin A/C is an important element of the cellular mechanotransduction pathway which bridges mechanical forces to the cell nucleus. In mesenchymal and non-epithelial tissues Lamin A/C is known to be regulated by ECM stiffness[20]. How Lamin A/C is regulated in epithelial tissues was not known. In this work, we show that Lamin A/C levels in epithelial tissues depend on apico-basal cell compression. We found that in many different epithelial tissues and under different genetic perturbations Lamin A/C levels can be reliably predicted solely based on apico-basal cell compression, suggesting that cell shape regulates Lamin A/C levels. We also showed that the nucleus flattens in response to apical-basal cell compression. It has been shown in recent studies on mesenchymal and non-epithelial tissues that the levels of Lamin A/C are higher when nuclei flatten on a stiff ECM. Recent studies in mesenchymal tissues have shown that tissue stiffness and cytoskeletal tension that lead to increased forces on the nucleus deform the nucleus[46–50]. In contrast, epithelial tissues alter cell shape by modulating cortical tension and properties of cell–cell junctions[36,51]. Our findings suggest that in epithelial tissues, Lamin A/C is regulated independent of tissue stiffness and ECM levels and that Lamin A/C levels depend on the flattening of the nucleus. Thus, nuclear deformation is a common feature of Lamin A/C regulation in both mesenchymal[49] and epithelial tissues, suggesting the at the scale of the nucleus similar mechanisms regulate the levels of Lamin A/C in both types of tissues. While we showed that LamC is regulated by apico-basal cell compression at a protein level, a recent study has shown that specific transcription factors could regulate the transcription of LamC[52].

In response to apico-basal cell compression, the nuclei of MDCK cells flatten and the nuclear surface area increases, resulting in stretching of the Lamin network. This stretching of Lamin network strongly correlates with dephosphorylation of Lamin A/C at serine 22, which could inhibit the degradation of Lamin A/C as observed previously in mesenchymal cells in culture. This could result in higher levels of Lamin A/C in apico-basally compressed cells. Interestingly Lamin A/C levels depend non-linearly on both apico-basal compression of the nucleus and on Lamin network stretching. We found that nuclear compression or Lamin network stretching had to exceed a threshold before Lamin A/C levels increased. Recent studies in cell lines and in Zebrafish embryos show a similar non-linear response of myosin II activity to a reduction of nuclear height. An increase in myosin II activity is observed only when the nucleus is compressed below a critical height, at which cytosolic phospholipase A2 (cPLA$_2$) was activated[43,44]. This phenomenon is called cellular proprioception, whereby the nucleus could sense and respond to changes in shape and size of a cell. In our work, pSer22 could act like a sensor of nuclear stretch to inhibit or activate degradation of Lamin A/C. Thus, cellular proprioception could be the mechanism through which

nuclei in epithelial tissues sense apico-basal cell compression and modulate the levels of Lamin A/C.

Interestingly, we also found that while cell morphology regulates Lamin A/C levels, the reverse is not true: changes in Lamin A/C levels do not influence the morphology of cells in *Drosophila* epithelial tissues. Even in Lamin C null mutants the morphology of the cells in the wing disc is not significantly different from wild-type, suggesting that altering Lamin A/C levels does not affect cytoskeletal organization in these cells. However, as previously described, the Lamin C null larvae do not survive beyond the late third instar stage of development[53,54]. This finding is in accordance with the role of Lamin A/C in protecting the genome against DNA damage[50,55] and in mediating nuclear mechanotransduction[56]. Thus, in LamC null larvae, hindered translocation of mechanotransducers like YAP[57] and MKL[58] or accumulation of DNA damage during larval stages could lead to lethality at later development.

Taken together, our results suggest that regulation of Lamin A/C could be the underlying mechanism coupling cell mechanics and nuclear mechanotransduction. The nature of this mechanism and how nuclear mechanotransduction could influence developmental process are major questions that our study raises for future investigations.

## Methods

**Fly genetics**. The fly experiments comply with all ethical regulations for animal testing and research. All flies were kept at normal cornmeal food and maintained at 25 °C unless otherwise specified. The fly stock, w-; DECADGFP fly was a kind gift from the Hong lab (Huang et al.). The fly stock, w-; UAS-CD8::Cherry; Dpp-GAL4, Gal80ts was a kind gift from Christian Dahmann. The fly stocks yw;vkg::GFP and w-;Ptc::Gal4,Gal80ts were obtained from the fly facility of Max Planck Institute of Molecular Cell Biology and Genetics, Dresden. The following fly stocks were obtained from Bloomington *Drosophila* Stock Center: w-; UAS-mmp2 (#58705),, w-; UAS-Lin (#7074), w-;UAS-CDC42$^{F89}$ (#6286), w-;UAS-CDC42$^{L89}$ (#6289), w-;UAS-yki$^{GFP}$(#28815), w-;UAS-yki$^{S168A}$ (#28818), w-;trol::GFP (#60214), w-;UAS-Stinger::GFP (#84277), w-;Df(2 R)trix/CyO (#1896), The following fly stocks were obtained from Vienna *Drosophila* Stock Center: w-;Laminin-A::GFP (#318155),. The fly stocks w-;UAS-LamC, w-; LamC$^{EX296}$/CyO were a kind gift from Dr. Lori Wallrath, University of Iowa. The fly stock w-;UAS-LamC$^{RNAi}$ was obtained from Fly stocks of National Institute of Genetics (#10119R-1). The fly stocks, w-;UAS-MMP2,Vkg::GFP and w-;CD8::Cherry; Ubx-GAL4 were generated in this work.

**Temperature Shift experiments**. Temperature shift was used to activate flies expressing Gal80ts. In the experiment, where Dpp-Gal4 is expressed under the control of Gal80ts, the larvae were grown at 18 °C till late third instar stage. As soon as first larvae come out of the food and start up crawling (about 10–11 days at 18 °C), the vial was shifted from 18 °C to 29 °C to activate the Gal80ts. The vial was kept at 29 °C for 24 h before dissecting and fixing the wing disc.

**Dissection of *Drosophila* tissues**. Late third instar larvae at up crawling stage were used for all experiments. The dissecting protocols for different *Drosophila* tissues are given below:

*Wing disc*. Larvae were selected and transferred to ice-cold 1X PBS for 5 min till the larvae were immobilized. Then the larvae were held by a pair of no. 55 forceps at one-third the body length from the anterior end and then the posterior part of the larvae is ripped with the help of another pair of no. 55 forceps. Then the mouth of the anterior part of the larvae is held by one pair of forceps and the larva was turned inside out revealing the imaginal discs. All the other tissues were removed leaving the just the wing imaginal disc attached to the body wall.

*Salivary gland*. The larvae were held at the middle by one pair of forceps and the mouth is held by another pair of forceps. The mouth was slowly pulled. As soon as the mouth parts start to come out, the forceps holding the middle of the larva was slowed released leading to the ejection of salivary glands by the fluid pressure inside the larva. The Salivary gland was cleaned off from the fat bodies attached and then collected in ice-cold 1X PBS for fixation.

*Trachea*. Larvae were held by the mouth and the posterior end of the larva was pinned into a PDMS pad. Then the anterior end was also pinned into the PDMS pad. Then the larva was ventrally dissected using fine forceps and the fat body, gut, and other organs were removed. Then the body wall was pinned to the sides and

the rest of unwanted body parts were removed leaving the trachea attached to body wall.

**Immunofluorescence staining of *Drosophila* tissues.** Different dissected tissues were fixed in 4% PFA for 20 min. Then the tissues were washed twice with PBX2 (1X PBS + 0.05% Triton X-100), 15 min for each wash. Then the tissues were washed in BBX250 (PBX2 + 1 mg/ml BSA + 5 mM NaCl) for 45 min. This was followed by overnight incubation at 4 °C in primary antibody diluted 1:50 in BBX (PBX2 + 1 mg/ml BSA). The following primary antibodies were used: mouse monoclonal antibody against Lamin C (DSHB, Cat No. LC28.26, 1:50 dilution), mouse monoclonal antibody against Lamin $DM_0$ (DSHB,Cat No. ADL67.10, 1:50 dilution), and rabbit monoclonal antibody against phosphor-Histone H3 (Sigma Aldrich, #06-570, 1:500 dilution). The primary antibody was washed twice with BBX, 20 min for each wash followed by washing in BBX + 4% normal goat serum once for 45 min. This was followed by incubation in secondary antibody either for 2–3 h at room temperature or overnight at 4 °C. The following secondary antibodies were used at 1:500 dilution: goat anti-mouse Alexa-Fluo647 (ThermoFisher, Cat No. A28181), goat anti-mouse Alexa-Fluo488 (Thermo Fischer, Cat No. A-21121). The secondary antibody was removed followed by washing twice with PBX2. Then the tissues were incubated in a mix of DAPI and either of the following Phalloidins at 1:50 dilution: Alexa-Fluo488-Phalloidin (ThermoFisher, Cat No. A12379), Alexa-Fluo568-Phalloidin (Thermo Fisher, Cat No. A12380), or Alexa-Fluo660 Phalloidin (ThermoFisher, Cat No. A22285). This was followed by washing twice with 1X PBS and the tissues were stored in PBS till the tissues were mounted. Before mounting, wing imaginal discs, salivary glands, and trachea were dissected from the body wall. Slides were prepared for mounting by creating a thin channel by using two strips of double-sided tape (Tesa). The tissue samples stored in 1X PBS were sucked in by a 100 μl pipette and transferred to the slide in between the two tape strips. The excess PBS was removed and the tissues were arranged. Then the sample was covered with 22 × 22 sized No.1 coverslip such that the coverslip sticks to the double-sided tape. Then Vectashield mounting medium (Vector laboratories) was added to one side of the coverslip and allowed to seep in by capillary effect. Once the entire sample was mounted, excess Vectashield was removed and the sides of the coverslip were sealed using transparent nail polish. The slides were then stored at 4 °C till they are imaged.

**Probing tissue stiffness by atomic force microscopy (AFM) indentation.** For AFM indentation measurements, a Nanowizard IV (JPK Instruments/Bruker, Berlin, Germany) mounted on top of a Zeiss light microscope (Axio Observer, 20× objective, Zeiss, Oberkochen, Germany) was used. Cantilevers (qp-BioAC, Nanosensors, Neuchatel, Switzerland or MLCT, Bruker Probes, Camarillo, USA) were calibrated prior experiments using built-in procedures of the SPM software (JPK Instruments). Experiments were performed at room temperature (18–20 °C). Tissues were immobilized using PLL (Sigma Aldrich, # P8920-100ML) or a thin layer of heptane glue in a glass-bottom dish (FluoroDish$^{TM}$, WPI, Sarasota, US). During the AFM indentation experiment, specific areas of interest on the tissue were selected for measurements with aid of the light microscope. The cantilever was then lowered at defined speed (5 μm/s) onto the tissue surface and retracted after reaching the force setpoint of 1nN. For each tissue, at least three different regions of interest were probed. Within each region, a 4 × 4 array of force-distance curves was acquired within an area of 25 × 25 μm²—for trachea, a higher number of smaller maps was chosen due to tissue geometry (10 regions, 2 × 2 points, 10 × 10 μm²). At least eight specimens were probed for each tissue. The resulting force-distance curves were transformed into force—versus—tip sample separation curves[1] and fitted with the Hertz/Sneddon model for a conical (qp-BioAC, angle 25°) or pyramidal indenter (MLCT, angle 18.5°), respectively[3,4], using the JPK analysis software (JPK DP, JPK Instruments). A Poisson ratio of 0.5 was used for the calculation of the apparent Young's modulus. Datasets were plotted using RStudio and compared using ANOVA package in MATLAB (Mathworks).

**Cell culture.** Madin Darby Canine Kidney (MDCK) cells were cultured in Minimum Essential Medium (MEM) (Gibco, Cat# 42360032), supplemented with non-essential amino acids (Gibco, Cat# 11140076) sodium pyruvate (Gibco,11360039) and 5% Fetal Bovine Serum (FBS) (Gibco). Cells were passaged using 0.25% Trypsin-EDTA (Life Technologies). Cells were cultured on 35 mm glass-bottom dishes (MatTek). A three-well cell culture insert (iBiDi, Cat # 80366) was inserted to form separate wells for low density and high-density culture, and the cover glass was treated with 100 μg/ml Collagen I PureCol solution (Advanced BioMatrix, Cat # 5005) and incubated for 2 h. The dishes were rinsed once with 1X PBS before seeding cells on them. For immunofluorescence, ~ 5700 cells/mm² were seeded for high-density culture, and 70 cells/mm² were seeded for low-density culture. For western blotting, $2 \times 10^6$ cells were cultured on a well of 12-well plate (high density) or on a 150 mm culture dish (low density) The cells were cultured for 24–48 h before fixation for immunofluorescence or before lysing for western blotting.

**Immunofluorescence staining of MDCK cells.** Cell culture medium was removed and cells were washed twice with 1X PBS. Then the cells were fixed in 4% Paraformaldehyde for 20 min. This was followed by washing twice with 1X PBS and then incubated in 0.5% Triton X-100 for 20 min. Then the cells were blocked for 20 min

in 1% BSA (made in 1X PBS). The cells were then incubated overnight with the primary antibodies diluted in 1% BSA made in 1X PBS. Then primary antibody was removed and the cells were washed twice with 1X PBS followed by incubating in 1% BSA for 20 min. Then cells were incubated for 2–3 h in the secondary antibody. The following primary antibodies were used—mouse monoclonal antibody against Lamin A (Thermo Fisher,Cat#MA1-06101, 1:100 dilution), mouse monoclonal antibody against Lamin B1 (Abcam, #ab8982, 1:100 dilution), rabbit polyclonal antibody against ZO-1 (ThermoFisher, Cat#40-2200, 1:500 dilution), monoclonal antibody against LamA phosphorylated at Ser22 (Cell signaling, Cat#2026S 1:125 dilution). The following secondary antibodies were used at 1:500 dilution in 1% BSA: goat anti-mouse Alexa 647Fluo (ThermoFisher, Cat No. A28181) and goat anti-rabbit Alexa 488Fluo (ThermoFisher, Cat No. A-27034).

**Testing conformation sensitivity of Lamin A/C antibodies.** Epitope masking has been previously shown to affect Lamin A/C staining in cultured cells[27]. Prior to quantification of Lamin A/C levels, the effect of epitope masking was tested. In *Drosophila* tissues, the intensity profile of LamC along the apico-basal axis of the nucleus was estimated. Similar intensity of LamC in both the apical and basal sides of the nucleus confirmed that epitope masking does not influence LamC staining in *Drosophila* tissues (Supplementary Fig 1). The antibody against mammalian LamA has been shown previously to be conformation insensitive. Further, intensity profiles along the apico-basal axis of the nucleus were used to confirm that this antibody is indeed conformation insensitive and does not influence LamA staining in MDCK cells (Supplementary Fig. 10).

**Western blotting.** MDCK cells were cultured for 24 h before proceeding for Western blotting. After 24 h, MDCK Cells were washed with 1X PBS and trypsinized with 0.25% Trypsin EDTA for 15 min at 37 °C. Trypsin was inhibited with 10% FBS medium and cells were pelleted following centrifugation at 800 rpm for 3 min. The cells pellet was lysed in 10 mM Tris-HCl with 1% SDS buffer. Total protein levels were measured using BCA assay (ThermoFisher, Cat No. 23225). Equal protein amount (60 μg) was reduced with 5× LDS buffer and loaded on 3-8% Tris-acetate gels (Novex Cat No. EA0375PK2), with Page ruler pre-stained ladder (ThermoFisher, Cat No. 26620). The gel was wet-transferred for 90 min at 20 V using NuPage transfer buffer and probed for mouse—Lamin A/C (ThermoFisher, Cat. No. MA1-06101, 1:100 dilution) and Rabbit-Actin (Abcam, Cat. No. 8227,1:1000 dilution) in 0.5% milk-TBST overnight at 4 °C. Lamin A/C and Actin were detected in 5% milk-TBST using donkey anti-Rabbit IgG IRdye 680 (Licor Cat No. 926-68073, 1:10000 dilution) and Donkey anti-Mouse IRdye 800 (Licor Cat No. 926-32212, 1:10,000 dilution), respectively. Western blot was imaged and quantified using Image Studio Lite (LI-COR Biotechnology, USA).

**Imaging**

*Imaging of Drosophila tissues.* Samples were imaged on an Olympus IX81 microscope equipped with a spinning disk module (Yokogawa) and back-illuminated EMCCD. Different imaging parameters were used for imaging different samples. Confocal z-stack of Lamin C immunostained samples were acquired using 40x silicon immersion objective with 0.47 μm z-spacing. Images of fluorescently tagged ECM proteins were imaged using 30x silicon immersion objective with 0.5 μm z-spacing. *Drosophila* wing disc expressing Nuclear GFP (Stinger-GFP), driven by Ubx-Gal4 was imaged using a 60x silicon immersion objective with 0.18 μm z-spacing. All images were acquired using a back-illuminated EMCCD camera (Andor Technology, iXon Ultra 888) with an exposure time of 100 ms and EM gain of 250.

*Imaging of MDCK cells.* MDCK cells were imaged on an Olympus IX81 microscope equipped with a spinning disk module (Yokogawa). Confocal z-stack of the cells were acquired using a 60x silicon immersion objective, with a z spacing of 0.28 μm, on back-illuminated EMCCD camera (Andor Technology, iXon Ultra 888) with an exposure time of 100 ms and EM gain of 250.

**Image analysis.** Analysis of images was performed using Fiji and MATLAB (Mathworks). We used separate analysis routines for analysis of images acquired for *Drosophila* tissues and MDCK cells.

*Analysis of Drosophila tissues.* Lamin C is present on the periphery of the cell nucleus. In order to estimate the levels of nuclear LaminC, the maximum projection algorithm in FIJI is used to project the Lamin C intensity from different planes over the nucleus in a z stack. Then using a custom-written program in MATLAB, the maximum projected plane is thresholded and multiplied with the maximum projected image to select pixels that belong to the nucleus. This is followed by calculating the mean intensity of Lamin C per pixel, which is a measure of level of Lamin C in the nucleus. Apical cell area and apico-basal height of the cells in *Drosophila* tissues were estimated using custom-written programs in MATLAB.

*Analysis of MDCK cells.* Cell morphology, nuclear morphology, and Lamin C levels in MDCK cells were estimated using custom-written program in MATLAB.

Fluorescence intensity of ZO-1 was used to estimate the apical cell area. ZO-1 network was segmented using Tissue Analyzer plugin in FIJI. The segmented image was inverted to estimate the apical cell area. Nucleus is segmented in 3-dimensions using MATLAB and the segmented nucleus is maximum projected along the $z$-axis and along the $y$-axis. The projection along the $z$-axis was used to estimate the cross-sectional area of the nucleus. An ellipse was fit to the projection of the nucleus along the $y$-axis and the minor axis of the ellipse was used as the height of the nucleus. The height of the cell was estimated from the height of the nucleus and the distance between the top of the nucleus and the top of the cell. In order to calculate the amount of Lamin A in the nucleus, the 3-D segmented nucleus was used as a mask and multiplied with the intensity image to select the pixels that belong to the nucleus. The total intensity of the masked 3-D nucleus is used as the total Lamin A intensity and the mean intensity per pixel is used as the level of Lamin A.

**Estimation of surface area strain of the nucleus**. The flattening of the nucleus was approximated as a transition from a prolate spheroid to an oblate spheroid. Since the nucleus is known to be fairly incompressible, it was assumed that the volume of the nucleus id conserved during this transition. The volume of a spheroid if given by

$$V_{spheroid} = \frac{4}{3}\pi a^2 c \tag{4}$$

where $a$ is the equatorial radius of the spheroid and $c$ is the radius in the orthogonal axis. Since for a given volume, sphere has the smallest surface area, a sphere was considered as the state with no surface area strain. Any deviation from the spherical shape leads to a surface strain. The radius of the sphere of volume $V_{spheroid}$ is

$$r_{sp} = \left(\frac{3V_{spheroid}}{4\pi}\right)^{1/3} \tag{5}$$

The surface area of this sphere is

$$S_{sp} = 4\pi r_{sp}^2 \tag{6}$$

The surface area of an oblate spheroid is

$$S_{ob} = 2\pi a^2 + \pi\frac{c^2}{e}\ln\left(\frac{1+e}{1-e}\right), \text{ where } e = \sqrt{1 - \frac{c^2}{a^2}} \tag{7}$$

The surface area of a prolate spheroid is given by

$$S_{pro} = 2\pi a^2\left(1 + \frac{c}{ae}\sin^{-1}e\right), \text{ where } e = \sqrt{1 - \frac{a^2}{c^2}} \tag{8}$$

Surface area strain is defined as the change in surface area relative to the surface area of the sphere of radius $r_{sp}$ and is given by,

$$\Delta S = \frac{S_{ob} - S_{sp}}{S_{sp}} \text{ or } \Delta S = \frac{S_{pro} - S_{sp}}{S_{sp}} \tag{9}$$

**Plotting and statistical analysis**. All graphs presented in the paper are plotted using R-Studio and the figures are prepared using Adobe Illustrator. Statistical analysis is performed using MATLAB. One-way ANOVA is used when multiple quantities are compared, whereas two-sided Student's $t$-test was used when two samples are compared. False discovery rate analysis was used to estimate the significance of correlations. One variable of the correlation is randomized iteratively for $10^5$ iterations and Pearson Correlation Coefficient (PCC) is estimated for each iteration. Then then the fraction of iterations with PCC better than the original PCC is estimated as the false discovery rate. We consider FDR < 0.05 as a significant correlation.

**Reporting summary**. Further information on research design is available in the Nature Research Reporting Summary linked to this article.

## Data availability
Relevant data and codes are available from the authors upon request. Source data are provided with this paper.

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

## Acknowledgements

We are grateful to Christian Dahmann for providing the *CD8-Cherry;Dpp-Gal4,Gal80^ts* fly line. We are grateful to Lori Wallrath, for the *UAS-LamC* and *LamC^EX296/CyO* fly lines. We thank the light microscopy facility at the MPI-CBG for help with the imaging. We thank Valentina Greco for valuable discussions and critical reading of the manuscript. We also thank Christian Dahmann, Shovamayee Maharana, and Suhrid Ghosh for critical review of the manuscript prior to submission. This work was supported by funding from the Deutsche Forschungsgemeinschaft: EA4/10-1, EA4/10-2 (N.A.D., K.V.I, S.E.). A.T is fellow of the Mildred Scheel Early Career Center Dresden P2 funded by the German Cancer Aid (Deutsche Krebshilfe).

## Author contributions

The project was conceived by K.V.I. Experiments were performed by K.V.I. The analysis was performed by K.V.I. AFM experiment was performed by A.T. with help from K.V.I. AFM data were analyzed by A.T. Western blotting was performed by S.A.Z. The first draft of the paper was written by K.V.I. with subsequent contributions by N.A.D., A.T., and F.J.

## Funding

## Competing interests

The authors declare no competing interests.
