## [Peer Review File · Nature Communications]

REVIEWER COMMENTS

Reviewer #1 (Remarks to the Author):

Iyer et al

This is an interesting manuscript that explores the regulation of nuclear Lamins, specifically the regulation of the mechano-responsive LaminA/C homolog LamC in *Drosophila*. The authors show that the levels of LamC are induced according to cell shape in epithelia, with very low expression in columnar cells, moderate expression in cuboidal cells, and highest expression in flattened squamous cells. Cell shape, rather than contact with the matrix, is the key determinant of LamC expression in *Drosophila*. Overall, this is an interesting set of observations, that goes beyond the current understanding of LamC regulation. I recommend publication after attention to the following minor comments.

Minor comments:

1. In their example of an epithelial cell in a fold (Fig 2d), the selected cell is far from the fold and quite columnar compared to the cuboidal cells close to the fold itself. So, it might be better to choose a more cuboidal example closer to the fold to highlight. The more columnar cells near the fold that express LamC may previously have been located within the fold and have recently moved out, hence retaining LamC through perdurance of the protein.
2. In the UAS.Lines experiment (Fig 4j), again the retained expression of LamC in the now-columnar peripodial epithelium may reflect perdurance of the LamC protein from when these cells were previously flat.
3. Fig 5 claims that cell shape regulates LamC *via* flattening of the nucleus. But the problem is that the columnar cells of the wing pouch also have a flat nucleus, just in a different orientation. So, their conclusion: "These results show that the observed changes in LamC in *Drosophila* tissues is due to deformation of the nucleus in response to apico-basal cell deformation" should be softened to say "These results suggest that the observed changes in LamC in *Drosophila* tissues may be due to deformation of the nucleus relative to the apical-basal axis of the cell in response to cell shape deformation". But it should be noted that no known mechanism exists to sense the orientation of the nucleus relative to the apical-basal axis of the cell. So it may be that LamC expression responds to other mechanisms that sense cell shape.
4. One mechanism for sensing cell shape deformation (mechanical strain) in *Drosophila* epithelia is the Hippo pathway. Can the authors test whether expression of UAS.Yorkie (wild-type or activated S168A mutant) can induce LamC expression?

Reviewer #2 (Remarks to the Author):

The manuscript by Iyer and coworkers describes differential laminA levels in different *Drosophila* tissues. They demonstrate that lamin A levels and nuclear shape scales with cell shape and not with ECM protein levels and that cell shape changes are sufficient to induce changes in lamin A, whereas lamin A levels do not regulate cell shape. The authors thus propose that apical compression of cells is sufficient to regulate lamin A levels. The mechanism of this effect is not addressed.

This is a clearly written, concise manuscript with interesting observations. The data is well presented and convincing. The main weaknesses of the manuscript are that most data is based on correlations and thus the mechanism of lamin A regulation remains elusive. In addition, the role of the matrix stiffness, which has been shown to regulate lamin A levels, has not been addressed. Finally, the observations of correlation of nuclear and cell shape has been reported previously and these studies

should be more clearly referenced.

Specific points:

1. The authors state that lamin A levels have been shown to scale with ECM concentration, but it would be more accurate to state that the levels scale with matrix rigidity, which relies on additional factors than just concentration, such as collagen crosslinking etc (see for example PMID: 24906154). Crosslinking is known to affect basement membrane mechanics without profoundly influencing its suprastructure. Matrix stiffness has a major contribution to cell shape and area, with a stiffer matrix promoting flatter cells, that could explain the correlation between lamin and Dcell (PMID: 30403543) as well as the role of cdc42 (PMID: 24393843). The authors should measure basement membrane stiffness for example using atomic force microscopy in the tissues as well as peroxidasin expression to exclude differences in basement membrane mechanics that could play an important role.

2. The finding that nuclear shape correlates well with cell shape and is consistent with a panel of previous work that shows that changes in cell boundaries are sufficient to alter nuclear shape independent of the lamina or the LINC complex (PMID: 26083918, PMID: 26287620, PMID: 26699462). The findings of the current study should be discussed in the context of this previous work.

Reviewer #3 (Remarks to the Author):

Observations that forces on the nucleus are transduced into lamin-A,C levels have slowly emerged, but what factors contribute to those forces and what tissues (in development or adult) exhibit such responses remain a matter of some speculation. This submission uses the powerful genetics and imaging possible with fly to document a relationship between lamin-A,C levels and cell & nuclear morphology in multiple epithelial tissues, including a couple of perturbations. The study is quantitative and compelling, but a number of concerns temper enthusiasm.

1. Quantitative immunostaining of LamC levels seems to assume that all LamC molecules are equally accessible to the antibody, but 'epitope masking' is a well known concern in the lamin field. The authors should do quantitative immunoblots or Mass Spec on the different tissues to support their quantitation.

2. The authors need to correct sentences in the main text and elsewhere similar to: "Since LamC levels in mesenchymal tissues are known to scale with ECM concentration 6,7." Reference-6 certainly includes non-mesenchymal tissues (e.g. lung, brain...), and shows LamC levels scale with tissue stiffness. Reference-7 is a review that extends that analysis to Col-1 in hearts as a contributor to stiffness in heart tissue in development and in adults. The authors should also describe References-8,23,30 as all pointing out the role of cytoskeletal tension, which relates to stiffness, increased forces on the nucleus, and increased lamin-A,C.

3. The statement "LamC is not regulated by ECM concentration in epithelial tissues" would require a complete accounting of any and all ECM, but the authors only quantify collagen IV for all tissues.

4. In the studies of MMP2 mediated degradation, does the tissue change stiffness locally or globally? In the studies of CDC42 overexpression: does cell proliferation change? does cell tension increase? does tissue stiffness increase?

5. Reference-23 should be cited as showing nuclear area tracks quantitatively with cell area, as used again in a different form here in Fig.5.

6. The authors conclude that "apico-basal cell compression regulates the concentration of Lamin A/C

by deforming the nucleus", but they should show this perturbs the cytoskeleton per References-8,23,30. Other perturbations would be more convincing

7. Regarding the MDCK studies: Compared to High density cells (HD), Low density (LD) cells are likely to be cycling with many more cells in S/G2 phase, which means more DNA and larger volume & area nuclei (up to 2-fold). Lamin-B levels therefore need to be quantified, especially since the average lamin-A levels differ by ~2-fold.

Reviewer #4 (Remarks to the Author):

In this article, Iyer and colleagues investigate the origin of the variability in Lamin A/C expression in epithelial cells, using drosophila embryos and cultured mammalian cells as model systems. They find that, independantly of the amount of extra-cellular matrix, or the tissue of origin, cells which are more flat have more Lamin A/C than cells that are more tubular. They also observe that this correlates with the shape of the nucleus, which is not surprising given the size of the nucleus, and they propose that nuclear shape might be related to the amount of nuclear Lamin A/C. They do not propose a clear function for that, since alteration in the amount of Lamin A/C has not effect in their particular assay. Overall, the observation and quantifications, as well as the range of perturbation experiments provided, together give a very nice support for the idea proposed by the authors, that cells can sense their own shape, and adapt their level of lamin A/C accordingly. I would thus recommend publication once the few concerns below are addressed:

Main concerns:

1) Many sentences in the article imply that a causal relationship was demonstrated between the shape of cells and the level of Lamin A/C in the nucleus. The causality was not demonstrated in my opinion, and even less with the nuclear shape. So the sentences should be changed to express the existence of a correlation but not of a causality.

IN the abstract: we reveal that apico basal cell compression regulates the concentration of Lamin A/C by deforming the nucleus. This is clearly not shown in the paper. replace 'regulates' by 'correlates with' and also remove 'by', but rather express a correlation with nuclear compression

This is also a problem on page 6: 'is due to deformation of the nucleus'. The results do not show that 'it is due to', but that it is 'correlated with'

Also on page 7: 'the regulation of Lamin A/C by apico-basal compression' this is also not right. It is a correlation, so not 'by'

Further down on the same page: 'we revealed a mechanism' - no mechanism was revealed, a correlation was revealed, which suggest a possible mechanism

Again same page: 'induce nuclear flattening and regulate Lamin A/C level' - it was not shown that it 'regulates', but that it 'correlates with'

2)There is a minor issue with the method used to detect the level of Lamin A/C. An article from the lab of Viola Vogel (Nature Materials 2015) showed that some Lamin A/C antibodies are sensitive to the conformation of the protein. The result is that when the nucleus is deformed (for exemple the flat face in cells plated on a flat surface), the antibody does not detect lamin A/C in the part in which the nuclear lamin A is stretched, although Lamin A/C is present and detected by other antibodies. Could the authors check that they did not use a conformation sensitive antibody? The simplest would be, at least for a set of control conditions, to use another antibody against Lamin A/C and check that the same difference is found between flat and tubular cells.

The authors could also perform western blots on specific tissues with more or less deformed cells (I am not sure whether this is actually possible or not in drosophila, but it is certainly possible with the MDCK cells plated at different densities).

Minor issues:

- Page 3: D is defined as the ln of the ratio, so maybe the ratio varies accross 4 orders of magnitude,

as said in the text, but not D!

- I think the author should add a few introductory paragraphs to their article. There is no introduction at all in the current manuscript!

- Why the authors do not show the correlation for single cells in tissues, but only averages of the two measures for a given tissue? They did not stain/image the lamin A/C and the cell shape in the same cells? It would give a more convincing result to have the correlation at the single cell level

- in the MDCK experiments, the relation between shape and lamin A/C is clearly not linear (figure 5n). It rather seem to have a sort of threshold of 'activation' for a certain level of deformation. This is very reminiscent of the phenomenon described by Lomakin et al. and Venturini et al (bioRxiv papers) where a specific nuclear deformation, which corresponds to full unfolding and stretching of the nuclear envelope, triggers a rapid cell response. A similar pathway might regulate Lamin A/C levels and give this sort of threshold response as a function of shape. The Roca-Cusachs lab also proposed that nuclear envelope stretching could induce nuclear pore opening upon nuclear deformation, changing the import of transcription factors like YAP. here again, it could cause such a non linear relation between deformation and response. The authors could comment on that and try to speculate on what sort of relation between shape and response could be expected depending on what is the sensor for deformation.

- The claim that ECM is not important (page 4) should be either made less strong or be better proven. The turnover of Lamin A/C in the nuclear lamina is known to be extremely slow (for mammalian cells, it is basically incorporated only at each division and almost does not turn over across an entire cell cycle of 24 hours). The authors could check that in the time of their ECM depletion, there could really be a change in Lamin A/C content in the nucleus. This might not be easy to prove. But then the authors should discuss more clearly the limitation of their experiment

RESPONSE TO REVIEWER COMMENTS

Reviewer's comments: **Italicized in black**

Author response: **Non-italicized in blue**

Reviewer #1 (Remarks to the Author):

Iyer et al

This is an interesting manuscript that explores the regulation of nuclear Lamins, specifically the regulation of the mechano-responsive LaminA/C homolog LamC in Drosophila. The authors show that the levels of LamC are induced according to cell shape in epithelia, with very low expression in columnar cells, moderate expression in cuboidal cells, and highest expression in flattened squamous cells. Cell shape, rather than contact with the matrix, is the key determinant of LamC expression in Drosophila. Overall, this is an interesting set of observations, that goes beyond the current understanding of LamC regulation. I recommend publication after attention to the following minor comments.

Minor comments:

1. In their example of an epithelial cell in a fold (Fig 2d), the selected cell is far from the fold and quite columnar compared to the cuboidal cells close to the fold itself. So, it might be better to choose a more cuboidal example closer to the fold to highlight. The more columnar cells near the fold that express LamC may previously have been located within the fold and have recently moved out, hence retaining LamC through perdurance of the protein.

Response: We thank the reviewer for pointing this out. As an example image, we have now selected a region well into the fold which have cuboidal cells that better corresponds to the actual data presented in Fig. 2g and 2h. We have made these changes in Fig 2d of the revised manuscript.

2. In the UAS.Lines experiment (Fig 4j), again the retained expression of LamC in the now-columnar peripodial epithelium may reflect perdurance of the LamC protein from when these cells were previously flat.

Response: We agree with the reviewer that a memory of a previous state could influence LamC levels in some cells. However, in our quantification, we have averaged data from different wing discs and cells in different positions in the peripodial membrane. So only a minority of cells might be influenced by this effect. Since we see a significant difference between *wt* and *Lin* overexpression cells, this implies that the effect of this memory is weak and influences only a small number of cells.

*3. Fig 5 claims that cell shape regulates LamC *via* flattening of the nucleus. But the problem is that the columnar cells of the wing pouch also have a flat nucleus, just in a different orientation. So, their conclusion: "These results show that the observed changes in LamC in Drosophila tissues is due to deformation of the nucleus in response to apico-basal cell deformation" should be softened to say "These results suggest that the observed changes in LamC in Drosophila tissues may be due to deformation of the nucleus relative to the apical-basal axis of the cell in response to cell shape deformation".*

Response: Deformations of the nucleus can lead to either prolate or oblate shapes, which both could appear flat in an image. We therefore agree that "flattening of the nucleus" is a misleading term. Figure 5 shows the identification of prolate and oblate nuclei, by computing the nuclear deformation index (D_{nuc}), which is positive for oblate nuclei of peripodial membrane and negative for prolate nuclei of the pouch. This figure also shows that deformations into an oblate shape influences LamC levels more than deformations into a prolate shape. This shows that deformation of the nucleus into an oblate spheroid by apico-basal cell compression leads to higher levels of LamC, but deformation into prolate spheroid (in pouch) does not lead to higher levels of LamC. Thus, a deformation of the nucleus into oblate shape results in an elevation of levels of LamC. We have now reworded the heading of Fig 5 as "Apico-basal cell compression regulates LamC via prolate to oblate deformation of the nucleus".

But it should be noted that no known mechanism exists to sense the orientation of the nucleus relative to the apical-basal axis of the cell. So it may be that LamC expression responds to other mechanisms that sense cell shape.

Response: Orientation relative to the apico-basal axis is not essential in our argument. Key is the change in nuclear shape. We suggest that nucleus is deformed as the cell is deformed and typically along the same axis. Two recently published papers Venturini et al and Lomakin et al, Science (2020)- show that as the cell is flattened, the nucleus also decreases its height through a phenomenon that they call cellular proprioception. Our observations are consistent with these studies. Thus, nuclear shape could depend on cell geometry. Thus, a specific mechanism for the nucleus to sense the deformation along the apico-basal axis is not needed.

4. One mechanism for sensing cell shape deformation (mechanical strain) in Drosophila epithelia is the Hippo pathway. Can the authors test whether expression of UAS.Yorkie (wild-type or activated S168A mutant) can induce LamC expression?

Response: This is a very good suggestion by the reviewer. We have overexpressed both the wild-type (*yki^{GFP}*) and activated form of yorkie (*yki^{S168A}*) in a stripe of cells in the wing disc using Dpp-Gal4. Interestingly we do not find any difference in the levels of LamC between the cells expressing *yki* and the cells not expressing *yki*. Thus, in the Drosophila wing disc, UAS-yorkie overexpression does not induce LamC expression. These data have now been included in the Supplementary Fig. 9 of the revised manuscript. We have also discussed this in the main text of the revised manuscript.

Reviewer #2 (Remarks to the Author):

The manuscript by Iyer and coworkers describes differential laminA levels in different Drosophila tissues. They demonstrate that lamin A levels and nuclear shape scales with cell shape and not with ECM protein levels and that cell shape changes are sufficient to induce changes in lamin A, whereas lamin A levels do not regulate cell shape. The authors thus propose that apical compression of cells is sufficient to regulate lamin A levels. The mechanism of this effect is not addressed.

This is a clearly written, concise manuscript with interesting observations. The data is well presented and convincing. The main weaknesses of the manuscript are that most data is based on correlations and thus the mechanism of lamin A regulation remains elusive. In addition, the role of the matrix stiffness, which has been shown to regulate lamin A levels, has not been addressed. Finally, the observations of correlation of nuclear and cell shape has been reported previously and these studies should be more clearly referenced.

Response: We thank the reviewer for the insightful comments on the manuscript. In the revised manuscript, we have included new data relating nuclear deformation and LamA phosphorylation. These data show that in nuclei that are flattened into oblate shape the Lamin network is stretched, and the phosphorylation of LamA at Ser22 is reduced. Since phosphorylation of LamA is known to activate its degradation, reduced phosphorylation could inhibit LamA degradation. This could result in higher levels of LamA in oblate nuclei. We propose that this could be a mechanism for cell shape dependent regulation of Lamin A/C levels in epithelial tissues. These data have now been included in Fig 7 and Supplementary Fig 12 of the revised manuscript. Furthermore, in the revised manuscript, we have now also studied the role of matrix and tissue stiffness in regulating Lamin A/C levels. We now also reference previous studies regarding correlation between cell shape and nuclear shape.

Specific points:

1. *The authors state that lamin A levels have been shown to scale with ECM concentration, but it would be more accurate to state that the levels scale with matrix rigidity, which relies on additional factors than just concentration, such as collagen crosslinking etc (see for example PMID: 24906154). Crosslinking is known to affect basement membrane mechanics without profoundly influencing its suprastructure. Matrix stiffness has a major contribution to cell shape and area, with a stiffer matrix promoting flatter cells, that could explain the correlation between lamin and Dcell (PMID: 30403543) as well as the role of cdc42 (PMID: 24393843). The authors should measure basement membrane stiffness for example using atomic force microscopy in the tissues as well as peroxidase expression to exclude differences in basement membrane mechanics that could play an important role.*

Response: We agree that our statement about previous work was not accurate and we have changed our statement in the revised manuscript (line 144), which is now more precise.

We agree with the reviewer that it is important to know the role of tissue stiffness in determination of LamC levels. As suggested by the reviewer, we therefore measured the stiffness of Drosophila epithelial tissues using atomic force microscopy (AFM) indentation experiments. We observed that the tissue stiffness of different regions of the wing disc are very similar, whereas the stiffness of salivary gland is much larger than that of the wing disc. When we correlated the tissue stiffness measured by AFM with LamC concentration, we did not find a significant correlation. These results are shown in Fig. 3 of the revised manuscript.

In order to understand the influence of ECM crosslinking as compared to ECM density, we now performed a knockdown of Peroxidase—a Collagen crosslinker in Drosophila and studied its effect on cell shapes and LamC levels. The morphology of the cells in the wing disc did not change. Moreover, LamC levels were also not affected by this knockdown. In the original manuscript we had shown that degradation of collagen by Matrix metalloprotease (MMP2) leads to folding of the wing pouch. Together, this shows that concentration of collagen is more important than its crosslinking for maintaining tissue shape in Drosophila wing discs. The new data on Peroxidase knockdown is shown in Supplementary Fig. 5d of the revised manuscript and also discussed in the main text.

2. *The finding that nuclear shape correlates well with cell shape and is consistent with a panel of previous work that shows that changes in cell boundaries are sufficient to alter nuclear shape independent of the lamina or the LINC complex (PMID: 26083918, PMID: 26287620, PMID: 26699462). The findings of the current study should be discussed in the context of this previous work.*

Response: We thank the reviewer for suggesting previous work showing that nuclear shape correlates with cell shape. We have now discussed these previous findings about correlation between cell shape and nuclear shape in our revised manuscript (line 298) and have cited the appropriate references.

Reviewer #3 (Remarks to the Author):

Observations that forces on the nucleus are transduced into lamin-A,C levels have slowly emerged, but what factors contribute to those forces and what tissues (in development or adult) exhibit such responses remain a matter of some speculation. This submission uses the powerful genetics and imaging possible with fly to document a relationship between lamin-A,C levels and cell & nuclear morphology in multiple epithelial tissues, including a couple of perturbations. The study is quantitative and compelling, but a number of concerns temper enthusiasm.

1. *Quantitative immunostaining of LamC levels seems to assume that all LamC molecules are equally accessible to the antibody, but 'epitope masking' is a well known concern in the lamin field. The authors should do quantitative immunoblots or Mass Spec on the different tissues to support their quantitation.*

Response: The reviewer has raised an important point. Epitope masking is a well-known concern in the Lamin A/C field because antibodies can be conformation sensitive. When performing staining of Lamin A/C in MDCK cells for the original manuscript, we had selected an antibody that was known to be conformation insensitive. We have now added the reference to this point in the revised manuscript (Reference no. 27 in the revised manuscript). This reference also shows that epitope masking can lead to differential staining of apical and basal sides of the nucleus even though the levels of LamA are the same. In order to test whether epitope masking affects our staining, we measured the intensity of Lamin A in the nuclei of MDCK cells and did not find any difference between the apical and basal side of the nucleus. For *Drosophila* cells, the LamC antibody that we have used is the only antibody available. In order to test whether epitope masking influence our staining, we measured the antibody staining in the apical and basal sides of the nucleus. We did not find any difference in intensity of LamC between the apical and basal sides of the nucleus, suggesting that our data is not affected by epitope masking.

In order to more directly address this issue, we have now performed immunoblots on MDCK cells plated at low and high density, as suggested by the reviewer. We observed a 2-fold difference in LamA levels in the low density culture as compared to the high density culture. This is consistent with our immunostainings of LamA and rules out that epitope masking leads to artefacts that affect our results. In *Drosophila* the levels of LamC varies within a single wing disc, making it difficult to perform quantitative immunoblots. We have included these data in the Supplementary Figure 1 and Supplementary Figure 11 of the revised manuscript. We have added a section regarding epitope masking in the methods section of the revised manuscript.

2. *The authors need to correct sentences in the main text and elsewhere similar to: "Since LamC levels in mesenchymal tissues are known to scale with ECM concentration 6,7." Reference-6 certainly includes non-mesenchymal tissues (e.g. lung, brain...), and shows LamC levels scale with tissue stiffness. Reference-7 is a review that extends that analysis to Col-1 in hearts as a contributor to stiffness in heart tissue in development and in adults. The authors should also describe References-8,23,30 as all pointing out the role of cytoskeletal tension, which relates to stiffness, increased forces on the nucleus, and increased lamin-A,C.*

Response: We agree that these papers report that LamC levels depend on tissue stiffness rather than ECM concentration. We have clarified this point in the revised manuscript. We have also measured tissue stiffness in the revised manuscript in order to clarify potential stiffness dependence of LamC. We find that in epithelial tissues LamC does not correlate with the tissue stiffness. We addressed this point in the response to the comment by the second reviewer (Comment no. 1). We also now mention that stiffness dependence of LamC was reported for other non-mesenchymal tissues. In the discussion of the revised manuscript, we now mention these references (Refs 46,49,50 in the revised manuscript) as pointing out the role of cytoskeletal tension on increased Lamin A/C levels.

3. *The statement "LamC is not regulated by ECM concentration in epithelial tissues" would require a complete accounting of any and all ECM, but the authors only quantify collagen IV for all tissues.*

Response: We agree that the statement was too general. Since understanding the role of ECM is an important point in our study, we have now quantified two additional ECM proteins – Laminin-A and Perlecan in wing discs, Salivary glands and Trachea of *Drosophila*. We find that in all these tissues, the levels of Collagen IV reported in the original manuscript are much higher than the levels of Laminin-A and Perlecan, suggesting that Collagen-IV is a major component of the ECM concentration. Moreover, as shown in the original manuscript, Collagen IV levels do not correlate with LamC levels and there was no evidence for correlation between LamC and Laminin-A and Perlecan. This suggests that LamC levels are not regulated by key ECM components. These data are now shown in Supplementary Figure 4 of the revised manuscript. We now state in the revised manuscript: "LamC is not regulated by the concentration of major ECM components in epithelial tissues" in line 163 of the revised manuscript.

4. *In the studies of MMP2 mediated degradation, does the tissue change stiffness locally or globally? In the studies of CDC42 overexpression: does cell proliferation change? does cell tension increase? does tissue stiffness increase?*

Response: We have shown in the original manuscript we have locally degraded the ECM by overexpressing MMP2 in a stripe of cells expressing Dpp-Gal4. We observe that Collagen-IV levels are significantly reduced and the tissue folds along this stripe of cells. This strongly suggests that tissue stiffness changes locally along this stripe. To address the question of the reviewer, we also looked at cell proliferation in CDC42 dominant negative (*CDC42^{L89}*) overexpression. We measured the cell proliferation by staining the wing discs using an antibody against phospho-histoneH3 that marks proliferating cells. We did not observe any difference in the fraction of proliferating cells in the region of *CDC42^{L89}* as compared to the wild-type cells, suggesting that in *CDC42^{L89}* overexpression cell proliferation does not change (shown in Supplementary Fig. 6b of the revised manuscript). Interestingly, cell shapes change and the tissue folds upon *CDC42^{L89}* overexpression. This suggests that cell mechanics changes locally giving rise to cell shape changes.

5. *Reference-23 should be cited as showing nuclear area tracks quantitatively with cell area, as used again in a different form here in Fig.5.*

Response: In the discussion of the revised manuscript, we have now cited reference 23 and few more references as showing that nuclear area tracks with cell area. The reference numbers are (46-50) in the revised manuscript.

6. *The authors conclude that "apico-basal cell compression regulates the concentration of Lamin A/C by deforming the nucleus", but they should show this perturbs the cytoskeleton per References-8,23,30. Other perturbations would be more convincing*

Response: We thank the reviewer for pointing this out. We have now shown that when the cells are compressed apico-basally by overexpressing a dominant negative form of CDC42, there is an accumulation of actin in these cells, suggesting that this perturbs the actin cytoskeleton, consistent with previous reports. This data is now shown in the Supplementary Fig 6a of the revised manuscript.

7. Regarding the MDCK studies: Compared to High density cells (HD), Low density (LD) cells are likely to be cycling with many more cells in S/G2 phase, which means more DNA and larger volume & area nuclei (up to 2-fold). Lamin-B levels therefore need to be quantified, especially since the average lamin-A levels differ by ~2-fold.

Response: As suggested by the reviewer, in the revised manuscript, we have quantified the levels of Lamin B1 in low density and high density cell cultures. We did not find any significant changes in the concentration of LamB1 in these two conditions. This shows that unlike LamA, LamB1 does not depend on packing density or cell morphology. These data are shown in Supplementary Fig 11 of the revised manuscript. Moreover we have also measured the intensity of the SUN domain protein, Klaroid (Koi) in the Drosophila wing disc. Even though it resides on the inner nuclear envelope, its intensity was similar in the different regions of the wing disc. This data is shown in the Supplementary Fig 2 of the revised manuscript. These data clearly show that the changes in levels of LamA/C that we observe in our experiments is indeed specific to Lamin A/C.

Reviewer #4 (Remarks to the Author):

In this article, Iyer and colleagues investigate the origin of the variability in Lamin A/C expression in epithelial cells, using drosophila embryos and cultured mammalian cells as model systems. They find that, independently of the amount of extra-cellular matrix, or the tissue of origin, cells which are more flat have more Lamin A/C than cells that are more tubular. They also observe that this correlates with the shape of the nucleus, which is not surprising given the size of the nucleus, and they propose that nuclear shape might be related to the amount of nuclear Lamin A/C. They do not propose a clear function for that, since alteration in the amount of Lamin A/C has not effect in their particular assay.

Overall, the observation and quantifications, as well as the range of perturbation experiments provided, together give a very nice support for the idea proposed by the authors, that cells can sense their own shape, and adapt their level of lamin A/C accordingly. I would thus recommend publication once the few concerns below are addressed:

Main concerns:

1) Many sentences in the article imply that a causal relationship was demonstrated between the shape of cells and the level of Lamin A/C in the nucleus. The causality was not demonstrated in my opinion, and even less with the nuclear shape. So the sentences should be changed to express the existence of a correlation but not of a causality.

Response: We agree with the reviewer that we do not have a definite proof of causality. We first showed that strong correlation exists between cell and nuclear shape and LamC levels. Based on this we formulated the hypothesis that there exists a causal relationship between cell shape and LamC levels. We then devised specific perturbations to test this hypothesis. In figure 4, we show that two different genetic perturbations which both alter cell shapes lead to corresponding changes in LamC levels, consistent with the hypothesis. To rule out the possibility that the correlation exists because LamC levels influence cell shapes, we directly perturbed LamC levels by either knocking down LamC in the peripodial membrane or overexpressing it in the pouch. In both these conditions, we did not observe any change in the cell shapes. Together, these data provide strong evidence that cell shape changes lead to changes in Lamin A/C levels.

We also agree with the reviewer that we had not demonstrated causality between nuclear shape and LamC levels in the original manuscript. To test whether a causal relationship exists, we have measured the nuclear shape upon CDC42^{L89} overexpression which we present in the revised manuscript. These data show that upon overexpression of CDC42^{L89} the shape of the nucleus changes from prolate to oblate and the levels of LamC increase (see Figure 5h-q).

Taken together, these experiments provide a strong evidence for a causal relationship between cell and nuclear shape and LamC levels, and also reveals a potential mechanism to Lamin A/C regulation in epithelial tissues. We have adjusted our statements in the revised manuscript to be more precise about what we can conclude from our experiments.

IN the abstract: we reveal that apico basal cell compression regulates the concentration of Lamin A/C by deforming the nucleus. This is clearly not shown in the paper. replace 'regulates' by 'correlates with' and also remove 'by', but rather express a correlation with nuclear compression

Response: As explained above, when using different genetic perturbation, we obtain results that go beyond correlations, but point to causalities. But we also agree that careful wording is important to be precise about our findings. We therefore now state in the abstract : “We provide compelling evidence that apico-basal cell compression regulates the levels of Lamin A/C by deforming the nucleus.”

This is also a problem on page 6: 'is due to deformation of the nucleus'. The results do not show that 'it is due to', but that it is 'correlated with'

Response: As mentioned above, we have now included in the revised manuscript, a study of the response of LamC to changes in nuclear shape, using CDC42 overexpression. The results show that the perturbation of nuclear shape leads to changes in LamC levels. This strongly suggests that a causal relationship exists. We therefore rewrite the sentence as “ These results suggest that the observed changes in LamC in Drosophila tissues are caused by prolate to oblate nuclear shape change in response to apico-basal cell compression”

Also on page 7: 'the regulation of Lamin A/C by apico-basal compression' this is also not right. It is a correlation, so not 'by'

Response: We have reworded the sentence. In the revised manuscript (line 316) we now state: “These results suggest that regulation of Lamin A/C by apico-basal compression of cells is evolutionarily conserved in epithelial tissues.”

Further down on the same page: 'we revealed a mechanism' - no mechanism was revealed, a correlation was revealed, which suggests a possible mechanism.

Response: We rephrased this sentence to: “In this work, we show that Lamin A/C levels in epithelial tissues depend on apico-basal cell compression.” Note also that in the revised manuscript, we have included new experiments to show that changing the shape of the nucleus from prolate to oblate reduces phosphorylation of Lamin A at Ser22 (pSer22), a marker for degradation of Lamin A. This reduction in pSer22 could lead to elevated levels of Lamin A/C in oblate nuclei of apico-basally compressed cells. Thus, our revised manuscript now reveals a potential mechanism of Lamin A/C regulation in epithelial tissues, which we now mention in the discussion of the revised manuscript. These data are shown in Fig. 7 and supplementary Fig. 12 of the revised manuscript.

Again same page: 'induce nuclear flattening and regulate Lamin A/C level' - it was not shown that it 'regulates', but that it 'correlates with'

Response: We have revised the discussion. In the revised manuscript (line 372), the statement now reads: “Our findings suggests that in epithelial tissues, Lamin A/C is regulated independent of tissue stiffness and ECM levels and that Lamin A/C levels depend on the flattening of the nucleus.”

2) There is a minor issue with the method used to detect the level of Lamin A/C. An article from the lab of Viola Vogel (Nature Materials 2015) showed that some Lamin A/C antibodies are sensitive to the conformation of the protein. The result is that when the nucleus is deformed (for example the flat face in cells plated on a flat surface), the antibody does not detect lamin A/C in the part in which the nuclear lamin A is stretched, although Lamin A/C is present and detected by other antibodies. Could the authors check that they did not use a conformation sensitive antibody? The simplest would be, at least for a set of control conditions, to use another antibody against Lamin A/C and check that the same difference is found between flat and tubular cells.

Response: This is an interesting point raised by the reviewer. When performing staining of Lamin A/C in MDCK cells, we were aware of the work from Viola Vogel’s lab. Hence we selected the antibody that is shown in the paper to be conformation insensitive. We have now included the reference to this article in the revised manuscript. In the Nature materials paper, it was shown that in a conformation sensitive antibody, the levels of Lamin A/C were significantly different in the apical and basal surfaces of the nucleus. However, when we measured the intensity of Lamin A in the nuclei of MDCK cells we did not find any difference between the apical and basal side of the nucleus, further validating that the antibody is not conformation sensitive. This data is now included in Supplementary Fig 11 of the revised manuscript. For Drosophila tissues, the LamC antibody that we have used is the only antibody available. In order to test whether this antibody is conformation sensitive, we measured the antibody staining in the apical and basal sides of the nucleus in both flat and spherical nucleus. We do not find any difference in intensity of LamC between the apical and basal sides of the nucleus. We have included these data Supplementary Fig 1 of the revised manuscript. Thus, these experiments show that conformation sensitivity of Lamin A/C antibody does not influence Lamin A/C staining in our study.

The authors could also perform western blots on specific tissues with more or less deformed cells (I am not sure whether this is actually possible or not in drosophila, but it is certainly possible with the MDCK cells plated at different densities).

Response : As suggested by the reviewer, we now performed western blots on MDCK cells plated at low and high densities. We found that the cells plated on low density have about twice the amount of LamA as compared to high density cells, consistent with the immunofluorescence staining of cells. This data is now included in the Supplementary Fig. 11 of the revised manuscript. This further validates that the levels of Lamin A/C observed in our study is not an artefact of conformation sensitivity of the antibody.

Minor issues

- Page 3: D is defined as the ln of the ratio, so maybe the ratio varies across 4 orders of magnitude, as said in the text, but not D!

Response: We thank the reviewer for pointing this out. We have changed this in the revised manuscript.

- I think the author should add a few introductory paragraphs to their article. There is no introduction at all in the current manuscript!

Response: We have now included a fully referenced introduction in the revised manuscript.

- Why the authors do not show the correlation for single cells in tissues, but only averages of the two measures for a given tissue? They did not stain/image the lamin A/C and the cell shape in the same cells? It would give a more convincing result to have the correlation at the single cell level

Response: This is an interesting point raised by the reviewer. We measured the cell shape index (D_{cell}) and LamC levels in the same cell in Drosophila tissues. We still observed a strong correlation between LamC and D_{cell} , but the correlation was non-linear at single cell level. This is very similar to the correlation between D_{cell} and LamA in MDCK cells. This data is now included in Fig 2 of the revised manuscript.

- in the MDCK experiments, the relation between shape and lamin A/C is clearly not linear (figure 5n). It rather seem to have a sort of threshold of 'activation' for a certain level of deformation. This is very reminiscent of the phenomenon described by Lomakin et al. and Venturini et al (bioRxiv papers) where a specific nuclear deformation, which corresponds to full unfolding and stretching of the nuclear envelope, triggers a rapid cell response. A similar pathway might regulate Lamin A/C levels and give this sort of threshold response as a function of shape. The Roca-Cusachs lab also proposed that nuclear envelope stretching could induce nuclear pore opening upon nuclear deformation, changing the import of transcription factors like YAP. here again, it could cause such a non linear relation between deformation and response. The authors could comment on that and try to speculate on what sort of relation between shape and response could be expected depending on what is the sensor for deformation.

Response: We thank the reviewer for pointing this out. We agree with the reviewer that the non-linear response of Lamin A/C to cell and nuclear shape is very similar to the phenomenon described in Venturini et al and Lomakin et al papers. In these papers, a threshold nuclear deformation triggers mechanotransduction in the nucleus by releasing cytosolic phospholipase A2 (cPLA2) by stretching of the nuclear envelope. In the revised manuscript, we have now shown that prolate to oblate shape change of the nucleus reduces the phosphorylation of Lamin A/C at Ser22. The shape change of the nucleus also leads to an increase in the surface area of the nucleus, suggesting that the lamin network stretches. Our data suggests that a threshold of stretching has to be exceeded for dephosphorylation to occur. Thus the non-linear response of Lamin A/C to nuclear shape could stem from the above threshold in dephosphorylation of Lamin A/C at Ser22, which could be a sensor of deformation. We have now included this data in Figure 7 and Supplementary Figure 11 of the revised manuscript. We have also discussed this in the discussions of the revised manuscript.

- The claim that ECM is not important (page 4) should be either made less strong or be better proven. The turnover of Lamin A/C in the nuclear lamina is known to be extremely slow (for mammalian cells, it is basically incorporated only at each division and almost does not turn over across an entire cell cycle of 24 hours). The authors could check that in the time of their ECM depletion, there could really be a change in Lamin A/C content in the nucleus. This might not be easy to prove. But then the authors should discuss more clearly the limitation of their experiment

Response: We agree with the reviewer that for complete turnover of lamin A/C occurs only during mitosis. A paper from the lab of Dennis Discher (Buxboim et al, Current Biology, 2014) shows that levels of Lamin A/C can change within few hours when plated on stiff substrate. This suggests that in our ECM depletion experiments, the time of ECM depletion (24 hours) is sufficient to observe a change in Lamin A/C levels. Thus, our result that ECM depletion does not lead to a change in LamC levels is not affected by the time needed to deplete the ECM.

REVIEWERS' COMMENTS

Reviewer #1 (Remarks to the Author):

I am satisfied with the revised manuscript, with one small exception:

In re-reading the manuscript, I notice that the authors did not cite the paper on regulation of LamC expression by the mechano-regulated ATF3 transcription factor in the *Drosophila* wing disc (Donohoe, Colin D.; Csordás, Gábor; Correia, Andreia; Jindra, Marek; Klein, Corinna; Habermann, Bianca; Uhlirova, Mirka. (2018) PLOS Genetics.)

<https://journals.plos.org/plosgenetics/article?id=10.1371/journal.pgen.1007241>

Here's a relevant figure showing that LamC is a bona fide target of Atf3:

https://plos.figshare.com/articles/figure/LamC_is_a_bona_fide_target_of_At3_/5939374/1

It would be important to add this citation to the manuscript, as it suggests a possible mechanism for their findings.

Reviewer #2 (Remarks to the Author):

The authors have done a great job revising the manuscript and have successfully addressed all concerns of this reviewer.

Reviewer #4 (Remarks to the Author):

In their revised manuscript, Iyer and colleagues included very nice new experiments which strengthen two major weaknesses of the previous version: they provide better proof of the causal link between nuclear deformation and regulation of the amount of lamin A/C, and they suggest a mechanism (nuclear envelope stretching leading to loss of phosphorylation of lamin A/C). Overall, they addressed most of my concerns and I would thus recommend publication.

They should give a detailed reading of their figures and figure legends for typos. For example, the Y axis of figure 4I is mislabelled (should be Dcell I think and not LMNA level). Also the figure legend of figure 6 has the wrong letters to refer to the figure panels. I did not check all in details, so there could be others.

A minor comment: the figures 4 and 5 seem very similar, because it is in fact quite obvious that the cell shape and nuclear shape should be strongly correlated, given the size of the nucleus compared to the size of the cell. So maybe it could be better to first show this correlation between the shape of both and then pull the results with Dcell and Dnucleus together. It is just a suggestion.

Response to Reviewers

Reviewer #1 (Remarks to the Author):

I am satisfied with the revised manuscript, with one small exception:

In re-reading the manuscript, I notice that the authors did not cite the paper on regulation of LamC expression by the mechano-regulated ATF3 transcription factor in the *Drosophila* wing disc (Donohoe, Colin D.; Csordás, Gábor; Correia, Andreia; Jindra, Marek; Klein, Corinna; Habermann, Bianca; Uhlirva, Mirka. (2018) PLOS Genetics.)

<https://journals.plos.org/plosgenetics/article?id=10.1371/journal.pgen.1007241>

Here's a relevant figure showing that LamC is a bona fide target of Atf3:

https://plos.figshare.com/articles/figure/LamC_is_a_bona_fide_target_of_At3_/5939374/1

It would be important to add this citation to the manuscript, as it suggests a possible mechanism for their findings.

Response: We have now cited this reference in the revised manuscript

Reviewer #2 (Remarks to the Author):

The authors have done a great job revising the manuscript and have successfully addressed all concerns of this reviewer.

Reviewer #4 (Remarks to the Author):

In their revised manuscript, Iyer and colleagues included very nice new experiments which strengthen two major weaknesses of the previous version: they provide better proof of the causal link between nuclear deformation and regulation of the amount of lamin A/C, and they suggest a mechanism (nuclear envelope stretching leading to loss of phosphorylation of lamin A/C). Overall, they addressed most of my concerns and I would thus recommend publication.

They should give a detailed reading of their figures and figure legends for typos. For example, the Y axis of figure 4I is mislabelled (should be Dcell I think and not LMNA level). Also the figure legend of figure 6 has the wrong letters to refer to the figure panels. I did not check all in details, so there could be others.

Response: We thank the reviewer for providing critical comments on the manuscript. We have now read the manuscript thoroughly and have now corrected all the error in the figures and figure legends.

A minor comment: the figures 4 and 5 seem very similar, because it is in fact quite obvious that the cell shape and nuclear shape should be strongly correlated, given the size of the nucleus compared to the size of the cell. So maybe it could be better to first show this correlation between the shape of both and then pull the results with D_{cell} and $D_{nucleus}$ together. It is just a suggestion.

Response: It is a good suggestion by the reviewer. But making this change alter the flow of the manuscript which will be rather difficult at this stage of the manuscript.